# Global and transcription-coupled repair of 8-oxoG is initiated by nucleotide excision repair proteins

Namrata Kumar[1,2], Arjan F. Theil [3], Vera Roginskaya[2,4], Yasmin Ali[4], Michael Calderon[5], Simon C. Watkins[5], Ryan P. Barnes[2,6], Patricia L. Opresko [2,6], Alex Pines[3], Hannes Lans [3], Wim Vermeulen [3] & Bennett Van Houten [1,2,4✉]

UV-DDB, consisting of subunits DDB1 and DDB2, recognizes UV-induced photoproducts during global genome nucleotide excision repair (GG-NER). We recently demonstrated a noncanonical role of UV-DDB in stimulating base excision repair (BER) which raised several questions about the timing of UV-DDB arrival at 8-oxoguanine (8-oxoG), and the dependency of UV-DDB on the recruitment of downstream BER and NER proteins. Using two different approaches to introduce 8-oxoG in cells, we show that DDB2 is recruited to 8-oxoG immediately after damage and colocalizes with 8-oxoG glycosylase (OGG1) at sites of repair. 8-oxoG removal and OGG1 recruitment is significantly reduced in the absence of DDB2. NER proteins, XPA and XPC, also accumulate at 8-oxoG. While XPC recruitment is dependent on DDB2, XPA recruitment is DDB2-independent and transcription-coupled. Finally, DDB2 accumulation at 8-oxoG induces local chromatin unfolding. We propose that DDB2-mediated chromatin decompaction facilitates the recruitment of downstream BER proteins to 8-oxoG lesions.

[1] Molecular Genetics and Developmental Biology graduate program, University of Pittsburgh School of Medicine, Pittsburgh, PA, USA. [2] UPMC Hillman Cancer Center, Pittsburgh, PA, USA. [3] Department of Molecular Genetics, Oncode Institute, Erasmus MC, University Medical Center Rotterdam, Dr. Molewaterplein 40, 3015 GD Rotterdam, The Netherlands. [4] Department of Pharmacology and Chemical Biology, University of Pittsburgh School of Medicine, Pittsburgh, PA, USA. [5] Center for Biologic Imaging, University of Pittsburgh, Pittsburgh, PA, USA. [6] Department of Environmental and Occupational Health, University of Pittsburgh Graduate School of Public Health, Pittsburgh, PA, USA. ✉email: vanhoutenb@upmc.edu

Oxidation of DNA can lead to a myriad of base lesions in the cell, including single-strand breaks and oxidized bases[1]. Due to its low redox potential, guanine is the most readily oxidized base[2] leading to the formation of 8-oxoguanine (8-oxoG). This modification is one of the most abundant oxidative lesions in the genome, with an estimated steady-state level of about 1–2 lesions/$10^6$ guanines[3–5]. 8-oxoG is pre-mutagenic and if unrepaired, can cause G:C to T:A transversions[6–8]. Accumulation of mutations can lead to genomic instability, which is associated with various maladies such as ageing, cancer and neurodegeneration[9,10].

In mammalian cells, 8-oxoG is repaired by the base excision repair (BER) pathway[11–14]. 8-oxoG is recognized by 8-oxoguanine glycosylase (OGG1), which removes the damaged base by breaking the glycosidic bond between the damaged base and sugar moiety, creating an abasic site. Biochemical studies have shown that OGG1 is a bifunctional glycosylase, with a weak AP (apurinic/apyrimidinic) lyase activity that cleaves the phosphate backbone and creates a single-strand break, leaving a free 5′ phosphate and a 3′-phospho-α, β-unsaturated aldehyde (3′-PUA)[15,16]. This repair intermediate is processed by AP endonuclease (APE1), leaving a 3′OH and a deoxyribose-5′-phosphate (dRP). DNA polymerase β (pol β) removes the dRP and fills the gap, while DNA ligase III seals the repair patch, completing the process. Cellular studies have suggested that the weak AP lyase activity of OGG1 might not function during BER, and instead APE1 cleaves the resulting abasic site[16,17].

Several biochemical studies, using purified OGG1 on reconstituted nucleosomes, have shown that OGG1 activity is severely inhibited when 8-oxoG is buried in the nucleosome[18]. Although, in some sequence contexts, lesions facing outward are more accessible for initiation of repair[19,20]. Therefore, one major question in the field is how glycosylases act on occluded lesions hidden in a sea of undamaged bases that are organized into a highly compact chromatin structure[21,22]. To this end, a number of chromatin remodelers and histone modifiers such as RSC, FACT, and ISWI have been suggested to help facilitate the repair of 8-oxoG, see reviews[23–26].

Interestingly, several studies have suggested the involvement of the nucleotide excision repair (NER) proteins in the repair of 8-oxoG, reviewed in[27–32]. NER is dedicated to the removal of bulky and helix-distorting lesions such as UV-induced photoproducts: 6-4 photoproducts (6-4PP) and cyclobutane pyrimidine dimers (CPD). Depending on the location of damage, NER is initiated through two sub-pathways: global-genome NER (GG-NER) and transcription-coupled NER (TC-NER)[33,34]. GG-NER is initiated by two protein complexes, UV-damaged DNA binding protein (UV-DDB) and XPC-RAD23B-CEN2. UV-DDB consists of the DNA-binding subunit DDB2 and the DDB1 subunit. Upon UV-induced DNA damage, UV-DDB, in complex with the CUL4A-RBX1 ubiquitin E3 ligase complex ($CRL^{DDB2}$), binds to the chromatin to ubiquitylate histones H2A, H3, and H4, making the lesion more accessible to downstream repair proteins in the NER pathway[35–37]. In addition, $CRL^{DDB2}$ also ubiquitylates XPC and DDB2 itself. TC-NER is initiated by the presence of a stalled RNA polymerase II (Pol II) at a lesion site on a transcribed strand, which is recognized by CSB, CSA and UVSSA[38]. Both sub-pathways converge at the damage verification step, which involves the recruitment of the TFIIH complex. XPA, RPA, and XPG stabilize the TFIIH complex at the DNA damage site. Damage removal is initiated by XPF-ERCC1, which makes an incision 5' to the lesion allowing DNA polymerase (δ/ ε/ κ) to begin repair synthesis triggering 3' incision by XPG, releasing the ~22–30 oligonucleotide excision product. Finally, DNA ligase I or III seals the newly synthesized repair patch.

NER and BER crosstalk has been suggested in several previous studies, reviewed in[27]. Cells deficient in XPC, XPA, CSB, and CSA exhibit delayed repair of 8-oxoG, after treatment with potassium bromate ($KBrO_3$), an oxidant that predominantly forms 8-oxoG lesions[39,40]. Using a system consisting of a photosensitizer (Ro 19-8022) plus 405 nm light to introduce predominantly 8-oxoG lesions[41], we demonstrated that XPC accumulated more efficiently to heterochromatic regions while CSB recruitment was targeted specifically to transcriptionally active regions[42,43]. Interestingly, contrasting models have been presented for the potential role of XPA in the removal of 8-oxoG[42,44]. While these studies suggest a role for NER proteins in facilitating 8-oxoG repair, a unified model of how NER and BER proteins work in synchrony is lacking.

Based on biochemical, single molecule and initial cell experiments, we have proposed a damage sensor role for UV-DDB in BER of 8-oxoG[45]. We showed that purified UV-DDB can stimulate activities of OGG1 and APE1 on DNA substrates containing abasic sites by 3-fold and 8-fold, respectively. Single-molecule DNA tightrope assays revealed that UV-DDB facilitates the displacement of OGG1 and APE1 from abasic sites. Finally, using a chemoptogenetic approach, consisting of a fluorogen activating protein (FAP) and a singlet oxygen specific photosensitizer[46], to introduce 8-oxoG lesions specifically at telomeres[47], we showed that DDB2 is recruited to 8-oxoG immediately after damage, preceding OGG1 recruitment[45]. However, these studies did not examine whether OGG1 processing of 8-oxoG is dependent on UV-DDB during BER in chromatin.

To gain mechanistic insights into how the NER proteins, DDB2, XPC, and XPA coordinate the processing of 8-oxoG in chromatin, we use two independent systems to introduce 8-oxoG in cells. This present study specifically addresses: (1) whether UV-DDB is required for recruitment of OGG1 to 8-oxoG; (2) the involvement of XPC and XPA in 8-oxoG repair; (3) the role of the $CRL^{DDB2}$ complex in dissociation of DDB2 from 8-oxoG sites; and (4) how DDB2 binding impacts the chromatin state at damage sites. We find that 8-oxoG removal is delayed in the absence of DDB2. Furthermore, the accumulation of XPC and OGG1 at 8-oxoG is DDB2-dependent. Moreover, we observe that binding of DDB2 to 8-oxoG at telomeres leads to telomeric chromatin decompaction, which allows for efficient recruitment of XPC and OGG1. We also identify that XPA facilitates 8-oxoG repair as part of the transcription-coupled repair (TCR) machinery, which is initiated when BER intermediates stall Pol II and block transcription. Based on these data, we propose a model for 8-oxoG processing that directly involves the NER proteins, DDB2, XPC, and XPA, where DDB2 binds 8-oxoG lesions to change the local chromatin environment facilitating the recruitment of downstream repair proteins.

## Results

**Repair of 8-oxoG is slower in the absence of DDB2.** Our previous study involving biochemical, single-molecule and cell experiments suggested that UV-DDB plays an important role in the processing of 8-oxoG during OGG1-mediated BER[45]. To determine whether DDB2 is directly involved in 8-oxoG repair, DDB2 and OGG1 were knocked down in U2OS cells and 8-oxoG levels were measured by immunofluorescence using 8-oxoG specific antibodies (Supplementary Fig. 1a, b), 48 h post transfection with siRNAs (Fig. 1a, b). We observed a significant increase of 1.3 and 1.8-fold in endogenous 8-oxoG levels in the absence of DDB2 and OGG1, respectively. These results suggest that DDB2 is involved in 8-oxoG processing in cells maintained at 5% $O_2$.

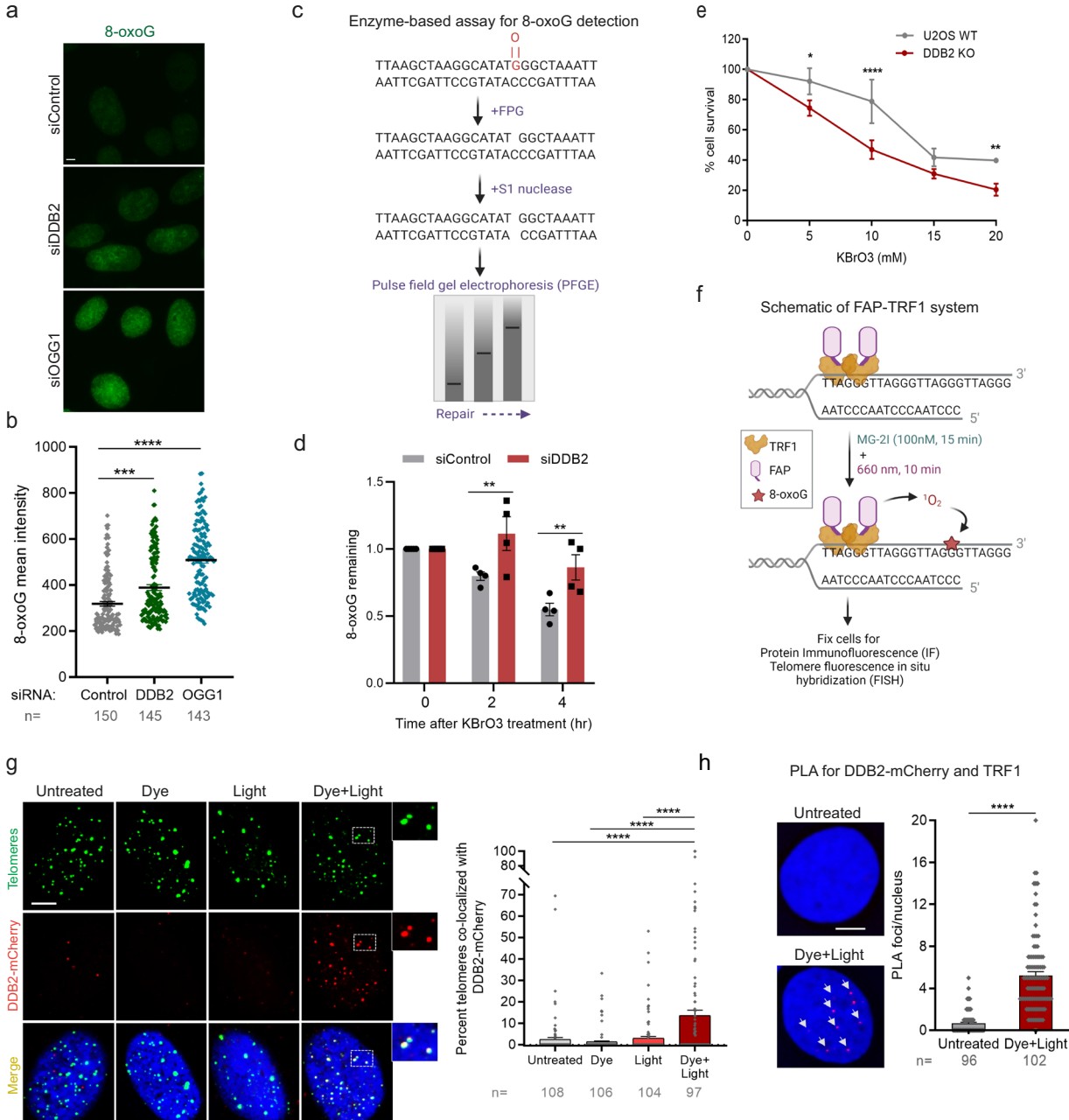

**Fig. 1 DDB2 facilitates 8-oxoG repair and is rapidly recruited to sites of 8-oxoG within telomeric DNA. a**, **b** Immunofluorescence and quantification of 8-oxoG in cells transfected with control, DDB2 or OGG1 siRNA. **c** Schematic of the repair enzyme-based assay for 8-oxoG quantification in DNA. Genomic DNA containing 8-oxoG is treated with FPG to convert 8-oxoG to one nucleotide gaps. Treating with S1 nuclease converts the gaps to double stranded breaks (DSBs). The cleaved DNA is subjected to pulse field gel electrophoresis (PFGE) to track repair, as damaged DNA migrates faster than repaired DNA. **d** Quantification of 8-oxoG repair in U2OS cells transfected with control or DDB2 siRNA and treated with KBrO3. **e** Clonogenic cell survival curves in U2OS WT and DDB2 knockout (KO) cells treated with a range of concentrations of KBrO3. **f** Schematic of dye plus light treatment. Cells stably expressing FAP-TRF1 were treated with dye (100 nM, 15 min) plus light (660 nm, 10 min) to introduce 8-oxoG lesions at telomeres. **g** (left) Recruitment of DDB2-mCherry to 8-oxoG sites at telomeres in untreated, dye alone, light alone, and dye plus light treated cells. (right) Percentage telomeres colocalized with DDB2-mCherry. **h** Proximity ligation assay (PLA) for DDB2-mCherry and TRF1 in untreated cells and cells treated with dye (100 nM, 15 min) plus light (660 nm, 10 min). Data (**a**, **b**, **d**, **g**, **h**) represent mean ± SEM from two to three independent experiments. "*n*" represents the number of cells scored for each condition. Data (**e**) shows one representative experiment (performed in triplicate) from three independent experiments, mean ± SD. One-way ANOVA (Sidak multiple comparison test) (**b**, **g**), Student's two-tailed Student's *t*-test (**h**) and two-way ANOVA (Sidak multiple comparison test) (**d**, **e**) were performed for statistical analysis: *$p < 0.05$, **$p < 0.01$, ****$p < 0.0001$, ns Not significant. Scale: 5 μm. Source data are provided as a Source Data file. (See also Supplementary Fig. 1 and 2).

In order to accurately measure the formation and repair of 8-oxoG, we adapted a protocol involving isolation of high molecule weight genomic DNA, digestion with Fapy DNA glycosylase (FPG) to convert 8-oxoG to one nucleotide gaps (Fig. 1c) and subsequent conversion to double-strand breaks (DSBs) with S1 nuclease, followed by pulse field gel electrophoresis (PFGE)[48,49]. In this experiment, cells were treated with $KBrO_3$ (40 mM, 1 h) and allowed to recover so repair could occur. The migration of the digested and non-digested genomic DNA was measured to calculate the mean DNA length. The relative amounts of 8-oxoG repaired over time was compared in cells transfected with either a control or DDB2 siRNA (Fig. 1d, Supplementary Fig. 2a-d). Repair of 8-oxoG was significantly delayed in DDB2 knockdown (KD) cells, indicating that DDB2 plays a direct role in BER of 8-oxoG.

To investigate whether loss of DDB2 had long term effects on cell growth after oxidative damage, we treated wildtype (WT) and DDB2 knockout (KO) cells (Supplementary Fig. 1h) with $KBrO_3$ before performing a colony formation assay. We found that cells deficient in DDB2 were more sensitive to oxidative DNA damage induced by $KBrO_3$ (Fig. 1e). As expected, OGG1 deficient cells were also sensitive to $KBrO_3$ treatment (Supplementary Fig. 1i). Taken together, these results indicate that DDB2 plays a critical role in 8-oxoG processing within genomic DNA.

**Robust recruitment of DDB2 to telomeric 8-oxoG lesions**. While $KBrO_3$ predominantly produces 8-oxoG lesions, to generate these lesions exclusively within a defined genomic region, we have recently developed chemoptogenetic approach to target 8-oxoG specifically at telomeres (Fig. 1f)[45]. This approach utilizes a fluorogen-activating protein (FAP) in combination with a photosensitizer dye, di-iodinated malachite green (MG-2I)[46]. Upon binding to FAP, the FAP plus MG-2I combination is excited by near-infrared wavelength (660 nm) to generate singlet oxygen[46]. Here, FAP is fused to a telomere binding protein, TTAGGG repeat binding factor 1 (TRF1), (FAP-TRF1)[45,47]. Singlet oxygen is highly reactive and short-lived and selectively forms 8-oxoG upon reaction with DNA[50,51]. Treatment with dye (MG-2I) plus light generates singlet oxygen that oxidizes guanines at telomeric DNA to form roughly 1-3 8-oxoG lesions/telomere[47].

Using this chemoptogenetic system, we previously showed that mCherry-tagged mouse DDB2 was recruited to 8-oxoG in human cells immediately after damage[45]. The recruitment of DDB2 preceded that of OGG1 suggesting that DDB2 may be the first responder in 8-oxoG recognition. Here, we confirmed and extended these results using a human DDB2-mCherry expressed in U2OS cells stably expressing FAP-TRF1 (U2OS-FAP-TRF1) and show that DDB2 is recruited to 8-oxoG after dye plus light treatment (Fig. 1g). As a parallel approach, we visualized the fluorescence of mNeon-DDB2 without using antibodies or DDB2-Flag in U2OS-FAP-TRF1 and RPE cells stably expressing FAP-TRF1 (RPE-FAP-TRF1) and observed robust recruitment of DDB2 after dye plus light treatment (Supplementary Fig. 1c, d). These results directly demonstrate that DDB2 recruitment to telomeric 8-oxoG is not cell type dependent. Moreover, N-terminal (mNeon-DDB2) or C-terminal tags (DDB2-mCherry, DDB2-Flag) result in similar recruitment frequencies (Supplementary Fig. 1c, d). In order to further validate that DDB2 is associated with telomeres after 8-oxoG damage, we utilized a proximity ligation assay (PLA) (Supplementary Fig. 1e–g). Antibodies against mCherry-tagged DDB2 and TRF1 were used and the PLA signal in untreated and dye plus light treated cells was examined. We observed a significant increase in PLA signal after dye plus light treatment (Fig. 1h), indicating that DDB2 is recruited to telomeres after 8-oxoG damage.

**XP-E K244E variant does not efficiently recognize 8-oxoG and UV photoproducts**. Mutations in DDB2 can cause xeroderma pigmentosum E (XP-E), a rare skin disorder characterized by extreme light sensitivity and increased risk of skin cancer[52]. We examined whether an XP-E variant K244E (Lys 244 to Glu) (Supplementary Fig. 1j) that is unable to bind specifically to UV-induced damage sites[53,54] (Supplementary Fig. 1k) can recognize 8-oxoG lesions in cells. We visualized the accumulation of WT or K244E DDB2-Flag at telomeric 8-oxoG in U2OS-FAP-TRF1 cells. Compared to WT, we observed a 2-fold reduction in DDB2 (K244E) binding to damaged telomeres (Supplementary Fig. 1l), indicating that the K244 residue is important for stabilization of DDB2 at sites of 8-oxoG damage.

**DDB2 is required for efficient OGG1 recruitment to 8-oxoG**. To evaluate the spatial and temporal association of DDB2 with OGG1 at sites of 8-oxoG damage, we employed PLA over a period of 3 h after dye plus light treatment. We observed a robust PLA signal from DDB2 and OGG1 immediately after dye plus light treatment that decreased to background levels by 3 h (Fig. 2a, b). These results strongly support the concept that DDB2 and OGG1 transiently associate during the processing of 8-oxoG. Biochemical and single-molecule results from our group have previously shown that UV-DDB stimulates the turnover of OGG1, and the two proteins transiently interact at abasic sites[45]. Strikingly, using IF we observed a higher accumulation of DDB2 at sites of damage in the absence of OGG1 at 30 min post dye plus light treatment (Fig. 2c, d, Supplementary Fig. 3a, c). By fitting these kinetic data to an exponential decay, we calculated an approximate 3-fold longer half-life ($t_{1/2}$) of DDB2 in the absence of OGG1 (Control siRNA = 30.65 min, OGG1 siRNA = 89.91 min). These data suggest that DDB2 continues to re-bind unrepaired lesions in the absence of OGG1.

While the abovementioned data supports the idea that DDB2 recruitment precedes OGG1, we wanted to examine whether DDB2 is absolutely required for OGG1 recruitment to 8-oxoG sites. To that end, we monitored the accumulation of OGG1 at damaged sites in the presence or absence of DDB2 (Supplementary Fig. 3b). Remarkably, when DDB2 was knocked down using siRNA, we observed a 3-fold reduction in OGG1 accumulation at both 30 min and an hour after dye plus light treatment (Fig. 2e, f). Consistent with these results, complete knockout of DDB2 resulted in a significant reduction of OGG1 accumulation at early times and longer retention at later times (Supplementary Fig. 3d, e). Together, these results establish that DDB2 is required for rapid and efficient recruitment and turnover of OGG1 at 8-oxoG sites.

**DDB2 recruits XPC to telomeric 8-oxoG, while XPA recruitment is transcription-coupled and independent of DDB2**. In GG-NER, UV-DDB facilitates the recruitment of XPC[55], which binds to the non-damaged strand and helps flip out the lesion on the opposite strand, facilitating the recruitment of the transcription factor TFIIH. Current models suggest that XPA is recruited simultaneously with TFIIH and is involved in both GG-NER and TC-NER[33]. To determine whether DDB2 mediates the recruitment of XPC and XPA to 8-oxoG, we examined the accumulation of GFP-tagged XPC or XPA over a period of 3 h after dye plus light treatment. In WT cells, we observed both XPC and XPA are recruited to 8-oxoG within 30 min post dye plus light treatment (Fig. 3a-d, Supplementary Fig. 4a, b), confirming the involvement of these proteins in 8-oxoG repair[39,40,42,44]. Interestingly, knocking out DDB2 decreased XPC accumulation by 3-fold (Fig. 3a, b). However, the recruitment of XPA to 8-oxoG damage was not affected even in the complete absence of DDB2 (Fig. 3c, d).

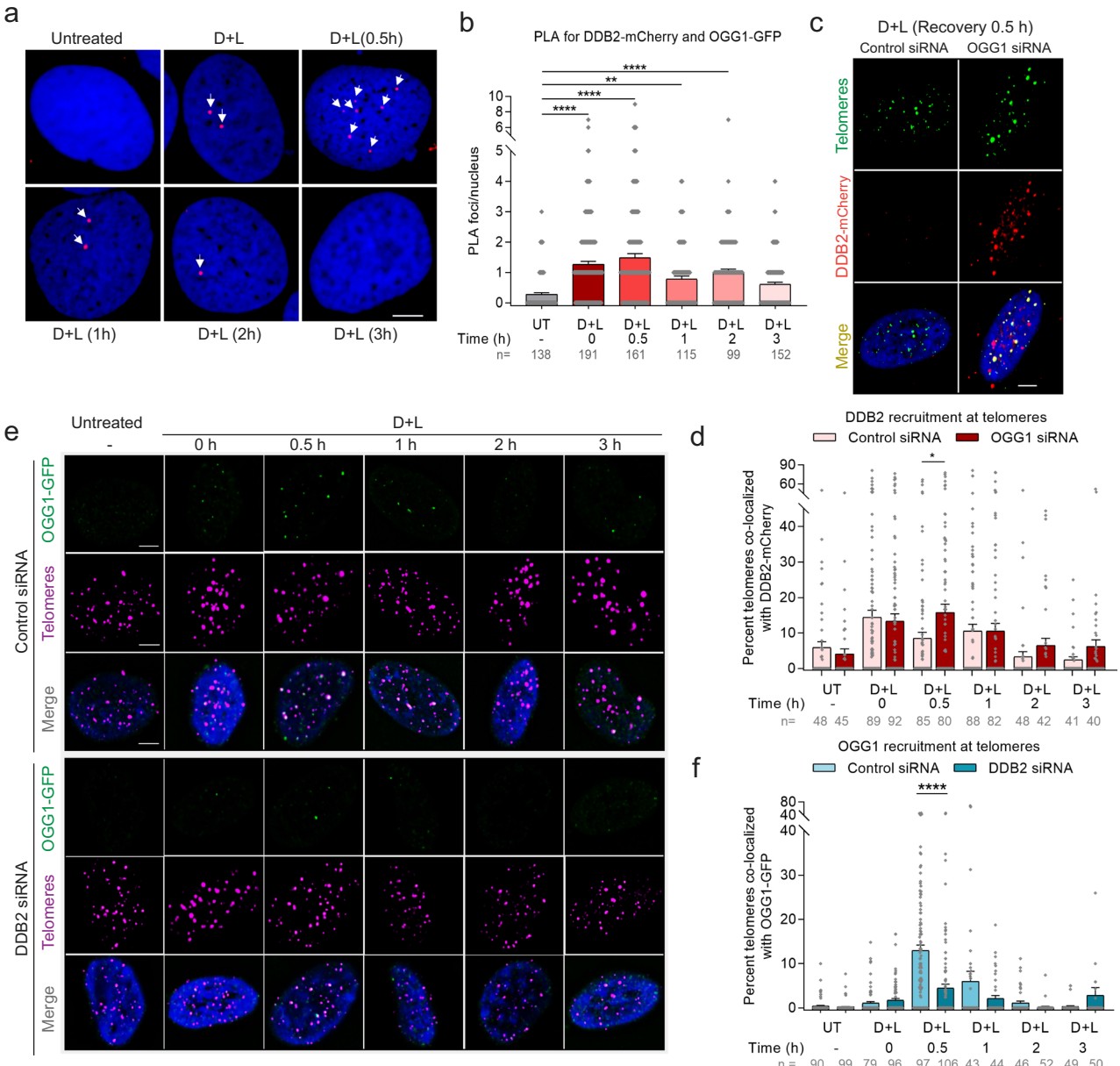

**Fig. 2 DDB2 is required for efficient OGG1 recruitment to 8-oxoG. a** DDB2-mCherry and OGG1-GFP associate at 8-oxoG sites as shown by PLA after dye (100 nM, 15 min) plus light (660 nm, 10 min) treatment, over a period of 3 h. Antibodies against mCherry and GFP were used. **b** Quantification of PLA. **c** Accumulation of DDB2-mCherry at telomeric 8-oxoG 30 min post dye plus light treatment in U2OS-FAP-TRF1 cells transfected with control or OGG1 siRNA. **d** Percent telomeres colocalized with DDB2-mCherry as shown in (c). **e** Recruitment of OGG1-GFP at damaged telomeres in cells transfected with control or DDB2 siRNA. **f** Percent telomeres colocalized with OGG1-GFP as shown in (**e**). Data (**a–f**) represents mean ± SEM from two independent experiments. "*n*" represents the number of cells scored for each condition. One-way ANOVA (Sidak multiple comparison test): *$p < 0.05$, **$p < 0.01$, ****$p < 0.0001$. Scale: 5 μm. Source data are provided as a Source Data file. (See also Supplementary Fig. 3).

These data suggest that XPC is recruited downstream of DDB2. Contrary to XPC, XPA appears to be recruited in a DDB2-independent repair pathway.

Spivak and colleagues have shown that cells lacking XPA were deficient in 8-oxoG repair in the transcribed strand[44]. To examine whether XPA is being recruited to sites of 8-oxoG damage through transcription-coupled repair (TCR) process, we pre-treated cells with transcription inhibitors, α-amanitin or THZ1, and analyzed the accumulation of XPC or XPA 30 min after treating with dye plus light. As expected, we saw no difference in XPC recruitment at 8-oxoG sites (Fig. 3e, f). Strikingly, we saw a 2-3-fold reduction in XPA accumulation in the presence of either transcription inhibitor (Fig. 3g, h), indicating that XPA

participates in TCR of 8-oxoG. The presence of TCR at 8-oxoG sites is noteworthy because 8-oxoG itself lacks transcription-blocking capacity[56]. However, it has been shown that BER intermediates (abasic sites and/or single-strand breaks) can efficiently block transcription[56,57]. Transcription of the C-rich strand "CCCTAA" at telomeres by Pol II gives rise to a class of long noncoding RNAs containing telomeric repeats (TERRA)[58]. While studies suggest TERRA plays a role in regulating telomere function and homeostasis, its mechanism of action is largely unknown[59]. The FAP-TRF1 system damages the G-rich strand of telomeres containing the "TTAGGG" repeat, which is the non-transcribed strand. Therefore, TCR of telomeric 8-oxoG is counterintuitive. We believe this could be due to two possible

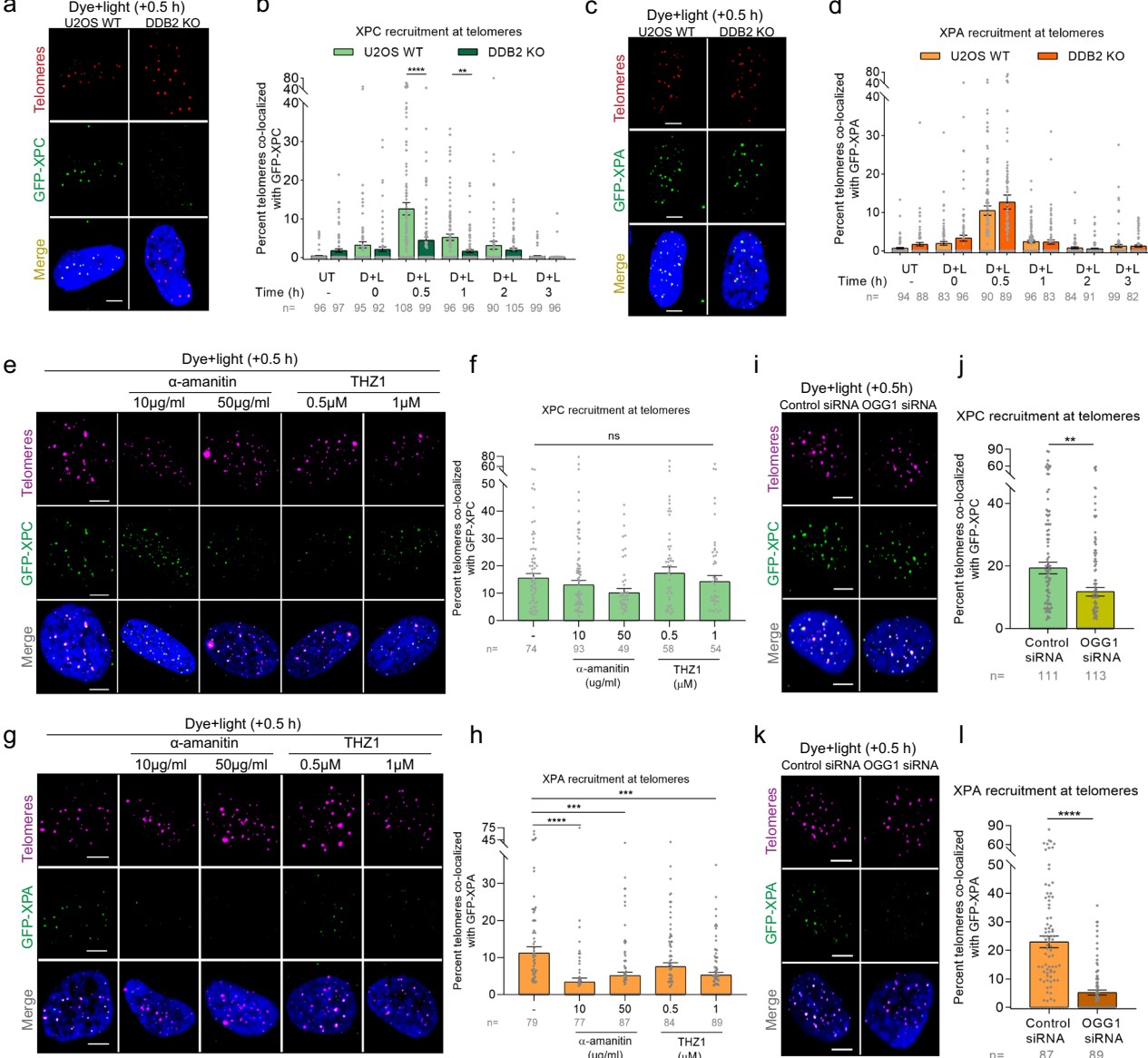

**Fig. 3 DDB2 recruits XPC to telomeric 8-oxoG, while XPA recruitment is transcription-coupled and independent of DDB2. a**, **c** Representative images showing recruitment of GFP-XPC (**a**) or GFP-XPA (**c**) to 8-oxoG at telomeres after dye (100 nM, 15 min) plus light (660 nm, 10 min) treatment in U2OS WT and DDB2 KO cells, 30 min post treatment. **b**, **d** Percentage telomeres colocalized with GFP-XPC (**b**) or GFP-XPA (**d**) after treatment, over a period of 3 h. **e**, **g** Representative images of GFP-XPC (**e**) or GFP-XPA (**g**) accumulation at damaged telomeres 30 min after dye plus light treatment in cells pretreated with transcription inhibitors α-amanitin and THZ1. **f**, **h** Quantification of e (**f**) and g (**h**). **i**, **j** Colocalization of GFP-XPC with telomeres after dye plus light treatment in U2OS-FAP-TRF1 cells transfected with control or OGG1 siRNA. **k**, **l** Colocalization of GFP-XPA with telomeres after dye plus light treatment in U2OS-FAP-TRF1 cells transfected with control or OGG1 siRNA. Data (**a**–**l**) represents mean ± SEM from two independent experiments. "*n*" represents the number of cells scored for each condition. One-way ANOVA (Sidak multiple comparison test) (**b**, **d**, **f**, **h**) and Student's two-tailed *t*-test (**j**, **l**): **$p < 0.01$; ***$p < 0.001$; ****$p < 0.0001$. Scale: 5 μm. Source data are provided as a Source Data file. (See also Supplementary Fig. 4).

reasons: (1) It has been shown that single-strand nicks in the non-transcribed strand favors the formation of R-loops, which involves the transcribed strand and efficiently blocks transcription, needing the TCR machinery to be recruited[60–62]; (2) U2OS cells maintain their telomeres through the recombination-mediated alternative lengthening of telomeres (ALT) pathway. ALT cells contain a "TCAGGG" variant repeat throughout the telomeres[63], therefore guanines are present in the complementary C-rich transcribed strand. The repair of this oxidized guanine might require TCR.

As mentioned earlier, Pol II stalls at BER intermediates, and formation of BER intermediates requires the action of OGG1

and/or APE1. To test whether XPA recruitment depends on OGG1-mediated processing of 8-oxoG, we knocked down OGG1 using siRNA (Supplementary Fig. 3a). We found that recruitment of XPA was decreased ~5-fold in the absence of OGG1 (Fig. 3k, l). Additionally, recruitment of XPA was dependent on CSB (Supplementary Fig. 4d), further validating that XPA is being recruited as part of the TCR machinery. In contrast, in OGG1 KD cells, XPC was still recruited to sites of 8-oxoG, although there was a slight 25% reduction (Fig. 3i, j). On the other hand, OGG1 recruitment was unaffected by the absence of XPC (Supplementary Fig. 4c). These data are consistent with previous published data showing that in human fibroblasts deficient for XPC,

recruitment of OGG1 is not affected, but dissociation of OGG1 is faster than in wildtype cells[42]. It is possible that XPC stabilization at 8-oxoG requires timely dissociation of DDB2 and subsequent recruitment of OGG1, similarly to our recent observation that timely dissociation of DDB2 and recruitment of the downstream GG-NER factor TFIIH stabilizes XPC binding to UV damage[64]. Moreover, it has been shown that XPC can stimulate OGG1 activity on 8-oxoG-containing duplex oligonucleotide by 3-fold[39]. In summary, our results indicate that 8-oxoG is processed at telomeres through two separate and distinct pathways: (1) a global repair pathway, involving GG-NER proteins, where DDB2 and XPC work together to enable OGG1 recruitment to 8-oxoG, and (2) a transcription-coupled repair pathway involving XPA, that is initiated when repair intermediates interfere with transcription.

**DDB2 binds sparse telomeric 8-oxoG lesions independently of the DDB1-Cul4A-RBX1 E3 ligase.** DDB2 was discovered as part of a heterodimeric complex, UV-DDB, consisting of DDB2 itself and the larger subunit DDB1[34]. UV-DDB forms a larger complex with the Cul4A-RBX1 ubiquitin E3 ligase (CRL$^{DDB2}$) and binds to UV damage to ubiquitylate histones and allow for chromatin relaxation and subsequent accessibility to downstream repair proteins, including XPC[35]. Interestingly, longer retention of UV-DDB at the damage site, either due to high affinity to the lesion or high lesion density, can obstruct downstream repair[64]. Therefore, timely removal of DDB2 is necessary for efficient repair. The CRL$^{DDB2}$ E3 ligase complex can auto-polyubiquitylate DDB2 to allow for its extraction from chromatin by the p97 segregase (VCP) and subsequent degradation by the 26S proteosome[64]. Other studies have shown that DDB2, in the absence of other factors, can cause chromatin decompaction and, together with ATP-dependent chromatin remodelers, alter the nucleosome structure around photoproducts after UV damage[65–67].

To evaluate the role of the CRL$^{DDB2}$ complex in 8-oxoG repair, we measured the accumulation of DDB2 after siRNA-mediated depletion of DDB1 or Cul4A (Supplementary Fig. 5a, b). We found that DDB2 binds to relatively sparse 8-oxoG sites (~1–3 per telomere[47]) even in the absence of CRL (Fig. 4a, b). To validate these results, we used PLA and confirmed that there was efficient recruitment of DDB2 to telomeric 8-oxoG, although it was slightly reduced in the absence of DDB1 (Supplementary Fig. 5c). It has been previously shown that loss of interaction with DDB1 renders DDB2 unstable in vivo[54]. Therefore, we measured DDB2 protein levels in cells transfected with DDB1 siRNA by western blot. We observed that knocking down DDB1 did not change the levels of endogenous DDB2 and led to a decrease in the overexpressed DDB2 protein levels, although this reduction did not reach statistical significance (Supplementary Fig. 5d).

To assess whether DDB2 dissociation from the damage site required the ubiquitylation action of DDB1 or Cul4A, we quantified the colocalization of DDB2-mCherry with GFP tagged DDB1 or Cul4A at 8-oxoG sites (Fig. 4c–f). We observed that DDB2 rapidly accumulated at sites of damage and dissociated by 30 min. On the other hand, we saw a significant accumulation of both DDB1 and Cul4A by 30 min. However, very little colocalization (<3%) was observed between DDB2 and DDB1 or Cul4A. Moreover, recruitment of DDB1 and Cul4A at 30 min was independent of DDB2 (Supplementary Fig. 5e, f). It is possible that the significant recruitment of DDB1 or Cul4A at 30 min post damage is due to TCR at repair sites, since DDB1 and Cul4A also associate with CSA (CRL$^{CSA}$) during TC-NER to ubiquitylate CSB[68]. Ubiquitylation and degradation of CSB have been shown to be indispensable for post TC-NER recovery of RNA synthesis[68,69].

The absence of colocalization between DDB2 and DDB1 or Cul4A could be due to low 8-oxoG density or differences in the repair of lesions in telomeres versus the bulk genome, since targeted damage to telomeric DNA only represents 0.02% of the genome. Therefore, much lower damage is being introduced after dye plus light treatment as compared to studies that have used high doses (10–60 J/m$^2$) of UVC to damage the entire genome. Additionally, the binding affinity of UV-DDB to 8-oxoG is ~5-fold lower than to CPDs[45], decreasing its retention time on the lesion, thus potentially eliminating the necessity for CRL$^{DDB2}$ mediated ubiquitylation and degradation. To that end, we looked at the total cellular DDB2 amounts in cells after dye plus light treatment. As expected, we saw no significant degradation of DDB2 after dye and light treatment (100 mW/cm$^2$, 10 or 20 min) or KBrO$_3$ treatment (40 mM, 1 h), but we saw as much as a 4-fold decrease in DDB2 levels 4 h after global UV damage (60 J/m$^2$) (Fig. 4g).

We recently demonstrated that DDB2 dissociation from UV damage is stimulated by recruitment of the downstream protein complex TFIIH, and longer retention on the damage site leads to CRL$^{DDB2}$ mediated DDB2 polyubiquitylation and degradation[64]. We, therefore, examined whether DDB2 and Cul4A colocalize at 8-oxoG sites in the absence of the downstream protein OGG1, 30 min post dye plus light treatment. Compared to WT cells, we observed a 2.5-fold increase in DDB2 and Cul4A colocalization at damaged telomeres when OGG1 was knocked down (Fig. 4h, i). As shown earlier (Fig. 3k, l), XPA is not recruited to telomeric 8-oxoG in the absence of OGG1, suggesting that 8-oxoG processing by OGG1 is required to form transcription blocking intermediates. To confirm that the Cul4A recruitment seen in OGG1 KD cells is not due to TCR, we treated OGG1 KD cells with α-amanitin and observed no effect on Cul4A recruitment (Supplementary Fig. 5g), suggesting that in the absence of OGG1, Cul4A is required for DDB2 dissociation from telomeric 8-oxoG. In total, these results indicate that at lower lesion densities and when OGG1 is present, DDB2 dissociation from 8-oxoG may not require CRL$^{DDB2}$ activity.

**DDB2 stimulates OGG1 recruitment to densely clustered 8-oxoG sites.** As shown in Fig. 4h, i, persistent binding of DDB2 to unrepaired 8-oxoG lesions results in the recruitment of the CRL complex. To validate DDB2's role in 8-oxoG recognition at higher lesion densities at non-telomeric sequences, we employed an independent approach using a photosensitizer (Ro 19-8022) in combination with 405 nm laser pulse[42,43] to locally induce 8-oxoG lesions at high density in specific sub-nuclear regions. We employed real-time live-cell imaging in three different cell lines stably expressing GFP-DDB2, OGG1-GFP or XPC-GFP and observed rapid recruitment (within a minute) of these proteins at 8-oxoG sites (Fig. 5a, b, Supplementary Fig. 6a, b). The recruitment of DDB2 or OGG1 was not observed when only single-strand breaks were introduced (Supplementary Fig. 6c, d).

In undamaged cells, the CRL$^{DDB2}$ complex is bound by the COP9 signalosome[70], which renders it inactive. Following UV damage, neddylation of Cul4A by NEDD8 makes CRL$^{DDB2}$ an active ubiquitin ligase. We used two different inhibitors to study this process: (1) NAEi, which inhibits neddylation keeping CRL$^{DDB2}$ inactive, and (2) CSN5i, which prevents deneddylation and keeps CRL$^{DDB2}$ hyperactive causing continual ubiquitylation and subsequent degradation of DDB2, even in the absence of UV damage (Fig. 5c, f). These inhibitors seem to be specific to CRL$^{DDB2}$ as CSA levels were unaffected. Using the NEDD8 inhibitor (NAEi) and keeping UV-DDB inactive significantly reduced OGG1 accumulation (Fig. 5d, e). Furthermore, when DDB2 is greatly depleted by the action of CSN5i, we observed a significant reduction of OGG1

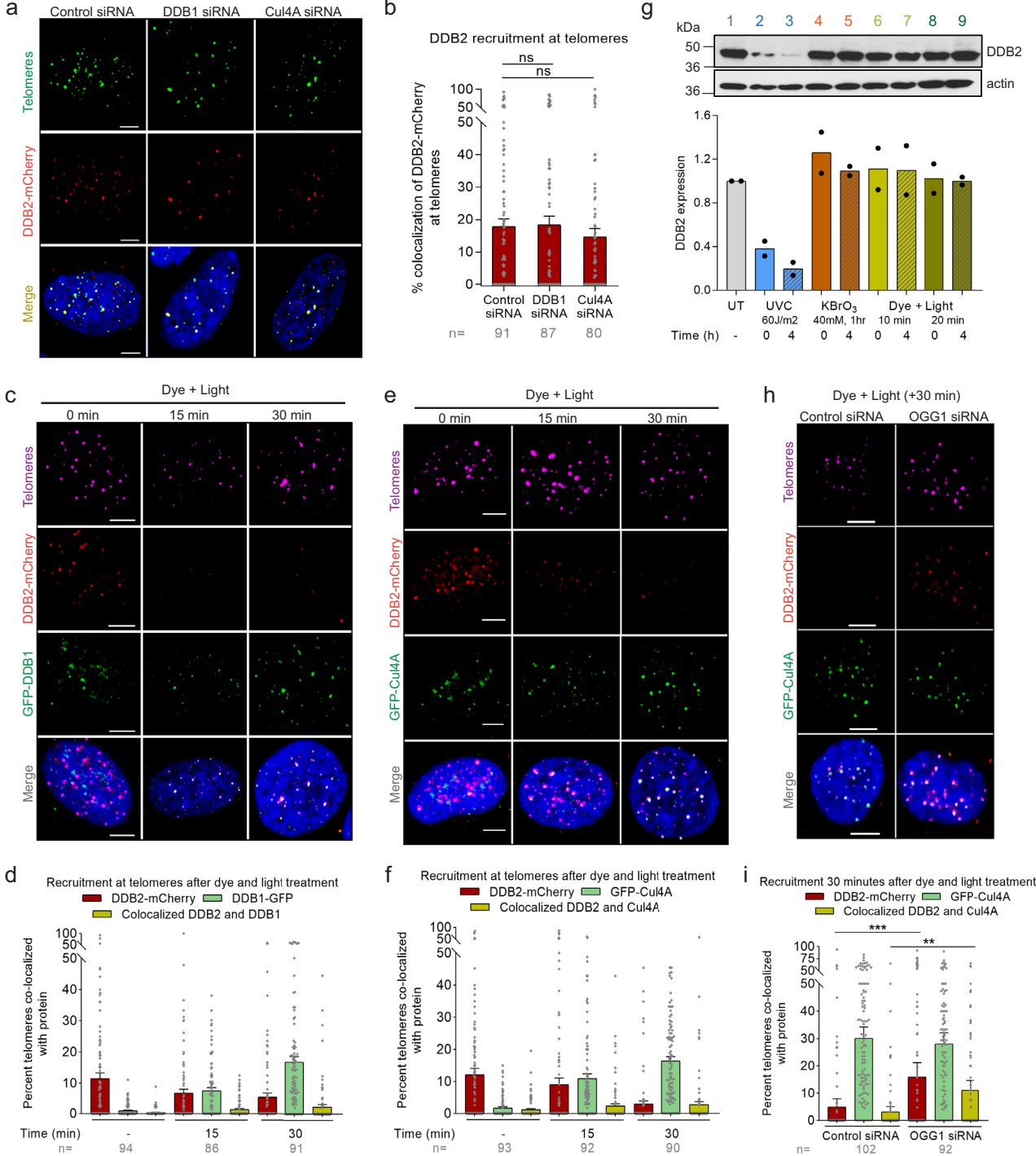

**Fig. 4 DDB2 binds sparse telomeric 8-oxoG independently of the DDB1-Cul4A-RBX1 E3 ligase. a** Representative images showing recruitment of DDB2-mCherry to telomeric 8-oxoG in cells transfected with control, DDB1 or Cul4A siRNA. **b** Quantification of a. **c, e** DDB2-mCherry and GFP-DDB1 (**c**) or DDB2-mCherry and GFP-Cul4A (**e**) accumulation at 8-oxoG sites after dye (100 nM, 15 min) plus light (660 nm, 10 min) treatment. **d, f** Quantification of c and e respectively. **g** Western blot for DDB2 in U2OS-FAP-TRF1 cells treated with UVC, potassium bromate (KBrO$_3$) or dye plus light at indicated doses. Independent experiments are represented by black circles. **h** Colocalization of DDB2-mCherry and GFP-Cul4A at damaged telomeres in U2OS-FAP-TRF1 cells transfected with control or OGG1 siRNA. **i** Quantification of h. Data (**a-h**) represents mean ± SEM from two independent experiments. 'n' represents the number of cells scored for each condition. One-way ANOVA (Sidak multiple comparison test) (**b, i**) was performed for statistical analysis: *$p < 0.05$, **$p < 0.01$, ***$p < 0.001$, ns Not significant. Scale: 5 μm. Source data are provided as a Source Data file. (See also Supplementary Fig. 5).

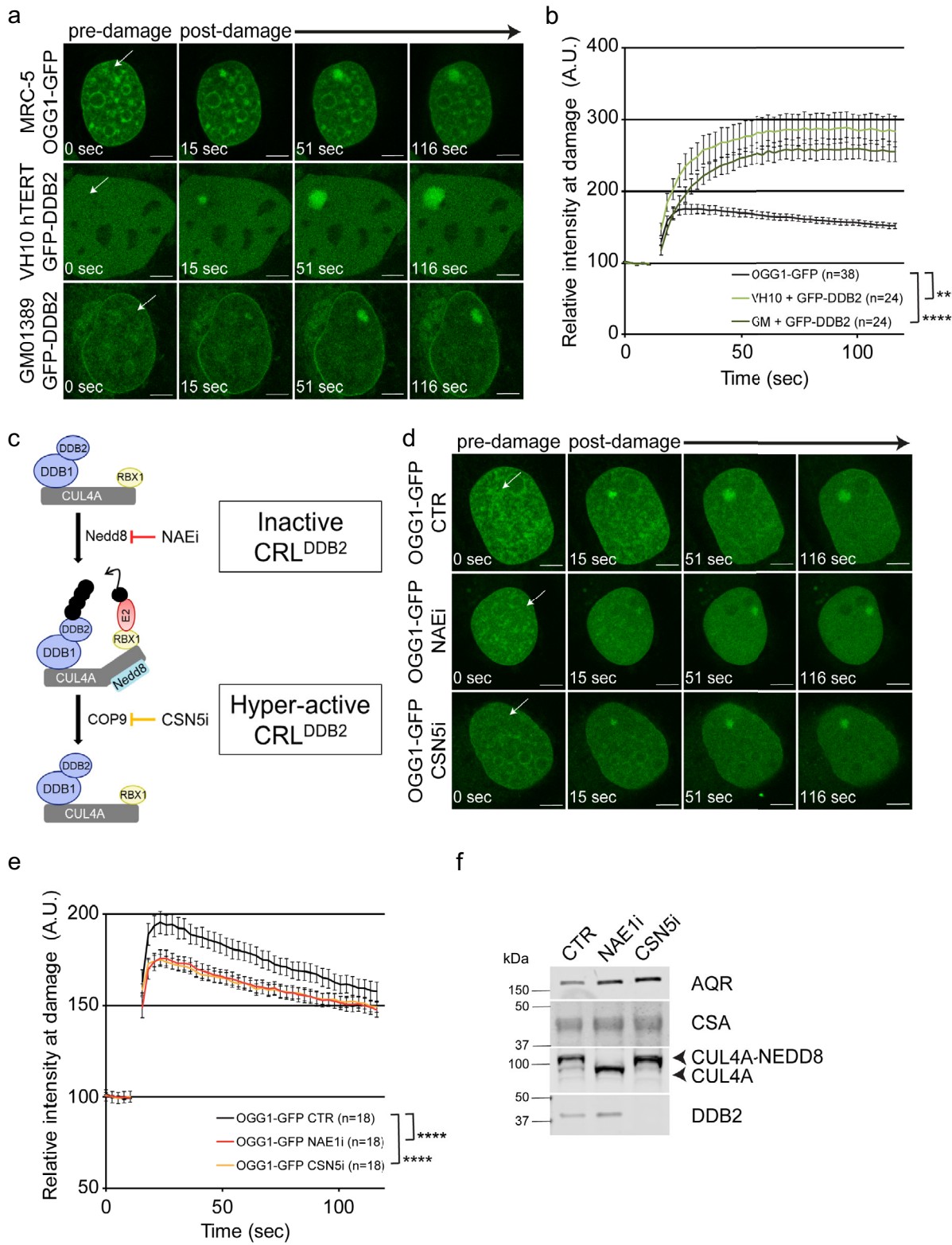

recruitment to 8-oxoG sites (Fig. 5d, e). Taken together these data suggest that DDB2 helps facilitate OGG1 recruitment to 8-oxoG sites irrespective of the genomic location. Moreover, either blocking CRL$^{DDB2}$ or ubiquitylating and degrading DDB2 reduces OGG1 recruitment to 8-oxoG, indicating the involvement of CRL$^{DDB2}$ during 8-oxoG repair when these lesions are at high densities in genomic DNA.

**DDB2 mediates chromatin decompaction at sites of telomeric 8-oxoG.** Intriguingly, we observed a gradual expansion of GFP-DDB2 and OGG1-GFP repair proteins at local 8-oxoG damaged sites after treatment with Ro 19-8022 and 405 nm light (Fig. 5a). As mentioned earlier, previous studies have shown a role for DDB2 in chromatin decompaction[65,66]. Moreover, in these studies, DDB2 was tethered to a Lac repressor (LacR) and expressed

**Fig. 5 DDB2 stimulates OGG1 recruitment to densely clustered 8-oxoG sites. a** Representative time-lapse pictures of OGG1-GFP and GFP-DDB2 accumulation at micro-irradiated (405 nm laser) sub-nuclear area, indicated by arrows, in the presence of 50 μM Ro 19-8022 photosensitizer.
**b** Quantification of accumulation kinetics of OGG1-GFP and GFP-DDB2 (as shown in a). **c** Schematic overview of the molecular interactions of DDB2 within the CUL4A-DDB1-RBX1 E3 ubiquitin ligase complex (CRL), which is required for the successive molecular interactions by ubiquitylation and subsequent DNA repair. The activation of CRL is mediated by covalent attachment of the ubiquitin-like activator NEDD8 on CUL4A and its proteolytic removal leads to the deactivation of ubiquitin ligase function. These crucial events can be fine-tuned by specific inhibitors MLN4924 (NAE1i) and SB-58-SN29 (CSN5i), acting on NEDD8-activating enzyme NAE1 and CSN5, respectively. **d** Representative time-lapse pictures of OGG1-GFP accumulation at micro-irradiated (405 nm laser) sub-nuclear area, indicated by arrows, in the presence of 10 μM Ro 19-8022 photosensitizer. Cells were pretreated with DMSO (CTR), NEDDylation inhibitor (NAE1i) or de-NEDDylation inhibitor (CSN5i) for 1.5 h. **e** Quantification of accumulation kinetics of OGG1-GFP (as shown in d).
**f** Immunoblot analysis for DDB2, CUL4A, CSA and AQR (loading control) in MRC-5 expressing OGG1-GFP. Cells were treated with inhibitors as indicated in d. Scale bars: 5 μm. Data were normalized to the background and represent mean ± SEM from three independent experiments. Total number of cells "n" measured are indicated in figure legends. ****$P < 0.001$, analyzed by ROC curve analysis. Source data are provided as a Source Data file. (See also Supplementary Fig. 6).

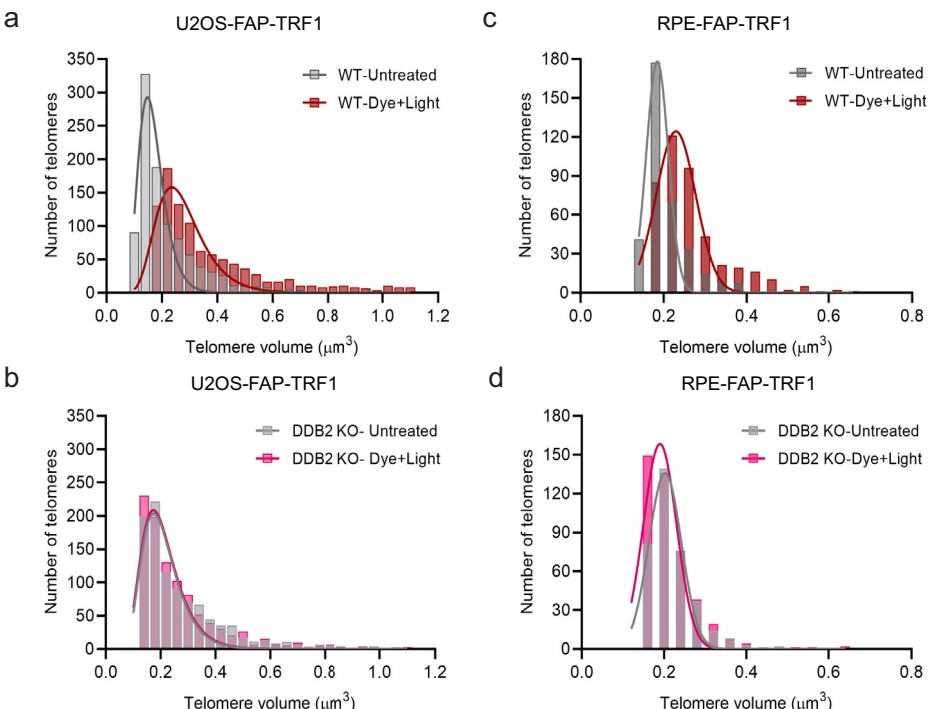

**Fig. 6 DDB2 mediates chromatin decompaction at sites of telomeric 8-oxoG. a**, **b** Distribution of the largest 20% telomeres in untreated and dye plus light treated U2OS-FAP-TRF1 WT and DDB2 KO cells. Cells were fixed 30 min post treatment. **c**, **d** Distribution of the largest 20% telomeres in untreated and dye plus light treated RPE-FAP-TRF1 WT and DDB2 KO cells. Cells were fixed 30 min post treatment. Source data are provided as a Source Data file. (See also Supplementary Fig. 7, Supplementary Movie 1 and 2).

in cells containing Lac operator (LacO) sites. Binding of DDB2-LacR to the LacO led to an expansion of the LacO area, suggesting that binding of DDB2 is necessary and sufficient for decompaction of chromatin. Based on these previous findings and our data, we asked whether binding of DDB2 to 8-oxoG lesions at telomeres impacted the local chromatin structure.

To address whether DDB2 binding to telomeric DNA causes telomere expansion, 8-oxoG was induced at telomeres and telomere 3D volumes in WT cells were measured using confocal imaging (Fig. 6a, Supplementary Fig. 7b, d, Supplementary Movie 1). These results indicated that the telomeric chromatin relaxes after 8-oxoG damage. Interestingly, this increase in telomere volumes was not observed when DDB2 was knocked out (Fig. 6b, Supplementary Fig. 7b, d, Supplementary Movie 1), indicating that DDB2 plays a critical role in local chromatin unfolding at the sites of 8-oxoG damage. U2OS cells maintain their telomeres through the alternative lengthening of telomeres (ALT) pathway, which is characterized by a heterogenous telomere length and telomere clustering after double-stranded

breaks (DSBs)[71]. To verify that the apparent telomere expansion we observed was not a result of ALT-associated telomere clustering, we measured the telomere volumes in a telomerase positive cell line, RPE-FAP-TRF1. We observed a significant increase in telomere volume in WT cells, but not in DDB2 KO cells (Fig. 6c, d, Supplementary Fig. 7a, c, e, Supplementary Movie 2). These data clearly demonstrate that DDB2 binds to 8-oxoG sites in the chromatin and mediates a local chromatin restructuring to allow downstream proteins to access the lesion. As DDB2 has no known chromatin remodeling activity, whether this decompaction is a direct result of DDB2 binding or through the recruitment of other chromatin remodelers remains to be investigated.

## Discussion

In this study, we used two complementary tools to introduce 8-oxoG sites at telomeric or local sub-nuclear regions and provided direct evidence for the involvement of several NER proteins

in 8-oxoG processing. We show that lack of the GG-NER protein DDB2 significantly delays 8-oxoG repair (Fig. 1d). DDB2 initiates 8-oxoG processing in chromatin immediately after damage is introduced (Fig. 1g, h). Furthermore, we observe that recruitment of XPC to 8-oxoG is facilitated by DDB2, suggesting that both UV-DDB and XPC act as early recognition factors in the repair of 8-oxoG. Strikingly, DDB2 knockdown by siRNA showed almost a complete inhibition of OGG1 recruitment at telomeres (Fig. 2f). Similarly, at locally induced 8-oxoG damage sites, a strong reduction of DDB2 by the COP9 signalosome deneddylation inhibitor, CSNi, which keeps the E3 ligase CRL$^{DDB2}$ in a hyperactive state, led to a decrease in OGG1 recruitment (Fig. 5e). XPA was also found to be recruited to sites of 8-oxoG damage at telomeres, and this recruitment is dependent upon OGG1 and transcription. We also found that in the absence of OGG1 or at high lesion density, UV-DDB was associated with Cul4A. Finally, we observed evidence for chromatin decompaction at 8-oxoG sites, which was dependent upon DBB2.

**Timely removal of DDB2 from unrepaired 8-oxoG lesions requires CRL$^{DDB2}$ mediated DDB2 dissociation.** UV-DDB, as part of the CRL$^{DDB2}$ complex, helps modify chromatin at sites of UV damage by ubiquitylating histones H2A, H3, and H4, and aids downstream NER[33]. Moreover, if UV-DDB remains bound to the lesion, CRL$^{DDB2}$ auto-polyubiquitylates DDB2 to allow for its dissociation and degradation. When 8-oxoG was produced at an apparent low density (roughly 1-3 per telomere) we found that recruitment and dissociation of DDB2 to 8-oxoG sites are independent of the CRL$^{DDB2}$ complex. We speculate that at low 8-oxoG density in telomeric DNA, DDB2 binding is transient, so DDB2 can dissociate without degradation. Interestingly, we observe repair protein foci even at these low-density lesions. It is possible that all three Gs of the TTAGGG sequence are converted to 8-oxoG after dye plus light treatment, which could explain the foci formation. Indeed, when we introduced a higher lesion density using the photosensitizer (Ro 19-8022) plus 405 nm laser illumination, we observed that OGG1 is not recruited effectively when CRL$^{DDB2}$ is inhibited. Recruitment of downstream NER proteins, such as TFIIH, can facilitate the dissociation of DDB2 from UV lesions[64]. Consistent with these findings, we saw a significant increase in DDB2 and Cul4A colocalization at telomeric 8-oxoG sites in the absence of OGG1, indicating that lesion density and location dictate whether DDB2 alone or in complex with DDB1-CUL4A-RBX are necessary for efficient OGG1 recruitment. Future experiments will focus on determining whether DDB2 dissociation in OGG1 KD cells is facilitated by CRL$^{DDB2}$ mediated DDB2 polyubiquitylation, subsequent action of VCP to extract ubiquitylated DDB2 from chromatin, and finally degradation by the 26 S proteosome.

**XPC and XPA participate in 8-oxoG processing through two independent sub-pathways.** Surprisingly, we observed both XPC and XPA recruitment to 8-oxoG. XPC recruitment was dependent upon DDB2. Why might XPC be recruited to sites of 8-oxoG processing, as these lesions are not expected to be processed by GG-NER? Biochemical experiments with purified XPC and OGG1 revealed that XPC can help turnover OGG1 at product inhibited abasic sites[39]. We have also shown with biochemical and single-molecule approaches that UV-DDB plays a similar role[45]. Furthermore, it was recently shown that repair of oxidative DNA damage was slower in XP-C cells compared to normal fibroblasts[72]. Future studies will be necessary to show that XPC and UV-DDB can work together to improve OGG1 access to damage and help turnover OGG1 during 8-oxoG processing, thereby stimulating the processing of 8-oxoG. Our present study

also clearly demonstrates that XPA recruitment is mediated through transcription-coupled repair. Previous studies have shown contrasting evidence for XPA's role in 8-oxoG repair[39,42,44], which could have been due to differences in experimental techniques and conditions. Here, we show that XPA is recruited to telomeric 8-oxoG as part of a transcription-coupled pathway when processing of 8-oxoG by OGG1 leads to transcription-blocking intermediates. Furthermore, we also observed TCR-linked recruitment of DDB1 and Cul4A suggesting an involvement of the CRL$^{CSA}$ complex in TCR of 8-oxoG. Future work will be necessary to determine if TC-NER recognition proteins, CSA and CSB, are recruited to actively transcribed regions at telomeric 8-oxoG to further define the interplay between GG-NER and TCR with BER.

**Chromatin structure defines the critical players required for 8-oxoG processing.** Chromatin structure can drastically affect the amount of oxidative DNA damage and repair in cells[73,74]. Specifically, it has been shown that heterochromatic regions are more susceptible to 8-oxoG damage[73], although this could be due to inefficient accumulation of BER proteins at heterochromatin compared to euchromatic regions[74]. Therefore, repair at heterochromatin may require additional factors including NER proteins. In this study, we observed a higher degree of DDB2 dependency on the recruitment of OGG1 when damage was introduced at telomeric chromatin versus at sub-nuclear genomic regions (Figs. 2f, 5e), suggesting that chromatin structure and lesion density play a key role in repair kinetics. Recent studies have established a chromatin decompaction role for DDB2 at sites of UV damage[65,66,75]. We demonstrate that when bound to 8-oxoG lesions, DDB2 facilitates chromatin expansion at sites of damage, as measured by increase in telomere volume. While DDB2 has been shown to lead to chromatin decompaction[65], it does not have any known chromatin remodeling properties. We, therefore, propose that DDB2 may mediate the change in chromatin state by recruiting other factors like chromatin remodelers. Chromatin remodelers and histone chaperones, such as RSC and FACT, have been shown to be involved in 8-oxoG repair[76]. Our DDB2 KO studies suggest that continued cellular absence of DDB2 activates compensatory pathways that facilitate less efficient recognition of 8-oxoG by OGG1. Future experiments are required to identify these additional factors. Furthermore, it is possible that DDB2 is required for 8-oxoG recognition in regions that are challenging for OGG1 to access. To that end, Thoma and colleagues have shown that UV-DDB can bind a lesion embedded in the nucleosome and even change the register of an occluded region by as much as three base pairs[75], suggesting a "pioneering repair factor" function for UV-DDB.

In NER, DDB2 is regulated by several post-translational modifications, including ubiquitylation, PARylation, and SUMOylation[35]. For example, it has been suggested that PARP1 mediated poly-ADP-ribosylation (PARylation) of DDB2 and subsequent recruitment of SWI/SNF chromatin remodeler, ALC1, facilitates repair of UV damage[67]. PARP1 also plays an important role downstream in BER by accumulating at BER intermediates (abasic sites/ single-strand breaks) and recruiting repair factors, XRCC1 and Pol β. More recently, ALC1 has also been shown to be required for BER[77,78]. To that end, it would be of importance to study the crosstalk between DDB2 and PARP1 at 8-oxoG sites undergoing repair.

In summary, our data support a model for how 8-oxoG lesions are processed at telomeres and other genomic regions, which consists of DDB2-dependent and -independent pathways (Fig. 7, Supplementary Movie 3). We propose that DDB2, alone at telomeres and as a CRL$^{DDB2}$ E3 ligase in other genomic regions,

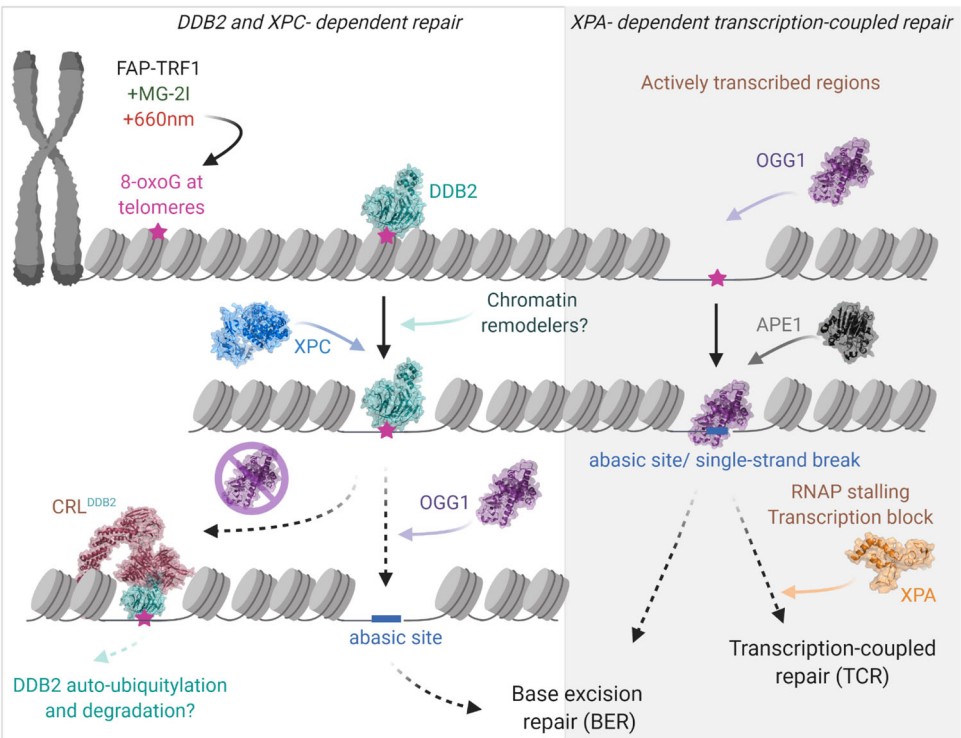

**Fig. 7 Unified working model: role of NER proteins in 8-oxoguanine repair.** Treatment of cells expressing FAP-TRF1 with dye (100 nM, 15 min) plus light (660 nm, 10 min) introduces 8-oxoG lesions at telomeres. In the DDB2-dependent repair pathway, DDB2 recognizes 8-oxoG lesions and facilitates chromatin relaxation through chromatin decompaction allowing the recruitment of XPC and OGG1 to the damage site. OGG1 recruitment facilitates the dissociation of DDB2. In the absence of downstream repair, DDB2 is retained longer at 8-oxoG sites requiring DDB1-Cul4A-RBX1 (CRL) mediated DDB2 dissociation. At actively transcribed regions, OGG1 can access the lesion independent of DDB2. 8-oxoG processing can lead to toxic BER intermediates that can act as a transcription block. Transcription-coupled repair (TCR) proteins, including XPA, participate in the repair of these BER intermediates. (See also Supplementary Movie 3).

binds 8-oxoG damage and facilitates local chromatin decompaction, stimulating damage recognition by XPC and OGG1. In contrast, if damage occurs in actively transcribed regions, where the chromatin structure is more relaxed, OGG1 may recognize damage independent of DDB2. Binding of OGG1 or processing of 8-oxoG by OGG1 and/or APE1 can stall Pol II, blocking transcription and requiring the recruitment of TC-NER proteins, including XPA. When the lesion density is high, re-binding of DDB2 to unrepaired lesions can inhibit downstream repair, requiring CRL$^{DDB2}$ mediated ubiquitylation and degradation of DDB2. Our study establishes a mechanistic role for NER proteins DDB2, XPC, and XPA in 8-oxoG processing. It remains to be investigated whether involvement of DDB2 and XPC in 8-oxoG repair is specific to heterochromatic and more condensed genomic regions that are tightly bound by nucleosomes, and thus in the absence of these GG-NER proteins would not efficiently be recognized by OGG1.

## Methods

**Cell lines**. The U2OS-FAP-TRF1 and RPE-FAP-TRF1 stable lines were obtained by transfecting pLVX-FAP-mCer-TRF1 plasmid in U2OS, and RPE-hTERT cells, respectively, and then selected in 500 μg/ml G418 (Gibco)[47]. Single cell cloning was used to select for cells with expression of FAP-mCer-TRF1 construct at telomeres. U2OS-FAP-TRF1 cells were cultured at 5% oxygen in Dulbecco's Modified Eagle Medium (DMEM) containing 4 g/l glucose (Gibco). RPE-FAP-TRF1 cells were cultured at 5% oxygen in Dulbecco's Modified Eagle Medium/ Nutrient Mixture F-12 (DMEM/F12 1:1) containing 2.438 g/l sodium bicarbonate (Gibco). Cells were supplemented with 10% fetal bovine serum (Gibco), 1×penicillin/streptavidin (Life Technologies) and 500 μg/ml G418.

SV40-immortalized MRC-5 cells stably expressing OGG1-GFP or XRCC1-YFP[43], hTERT-immortalized human fibroblasts VH10 stably expressing GFP-DDB2[67], hTERT-immortalized fibroblasts GM01389 (DDB2-deficient)[79] stably expressing GFP-DDB2 and XPC-deficient XP4PA-SV expressing XPC-EGFP[80]

were cultured at 37 °C in a humidified atmosphere with 5% CO2 in a 1:1 mixture of DMEM (Gibco, 41699-052) and Ham's F10 (Lonza, BE02-014F) supplemented with 10% fetal calf serum (FBS, FBS-12A) and 1% penicillin-streptomycin (Sigma, P0781).

**Knockout (KO) and knockdown (KD) cell line generation**. DDB2 knockout cells were generated in U2OS-FAP-TRF1 and RPE-FAP-TRF1 cells. HEK293T cells were co-transfected with a lentiviral construct expressing SpCas9 and a guide RNA targeting DDB2 exon 1 (Genscript DDB2 CRISPR Guide RNA1, CCGA-GATTGTATTACGCCCC), along with the Sigma CRISPR & MISSION® Lentiviral Packaging Mix (Sigma, SHP002). Briefly, $2.5 \times 10^5$ HEK293T cells (ATCC) were seeded in a 6-well plate. The next day 500 ng of the lentiviral vector, 4.6 μl of the lentiviral packaging mix and 2.7 μl of FuGENE 6 transfection reagent was incubated in 30.3 μl of OptiMEM. After incubation at room temperature for 15 min, the mix was added dropwise to each well containing 2 ml serum-free DMEM. The cells were incubated for 24 h at 37 °C and 5% oxygen. Next day, the media was replaced with 2 ml of fresh complete DMEM. Between 36–48 h post transfection, the supernatant was collected and filtered through a 0.2 μm filter. Fresh 2 ml complete DMEM was added to the HEK293T cells for the second harvest. The first harvest was added with 2 μl 10 mg/ml polybrene (Millipore #TR-1003-G) to the U2OS-FAP-TRF1 cells plated in a 6-well plate and incubated at 37 °C. The procedure was repeated for the second lentiviral harvest, between 60–72 h post transfection. The cells were then incubated overnight at 37 °C and 5% oxygen. Next day, fresh media was added, and cells were allowed to recover for 6–8 h. Cells were then selected with 1.5 μg/ml puromycin for 2 days in a 6 cm dish. Cells were then moved to a 75 cm² flask under selective pressure for an additional 2 days before harvesting for protein extraction and single cell cloning without puromycin. The clones were tested for DDB2 expression by western blot (abcam #ab181136) and immunofluorescence (abcam #ab51017). Clone 10 and Clone 37 were used for U2OS-FAP-TRF1 and RPE-FAP-TRF1, respectively.

**Plasmids**. mNeon-DDB2 was made by Gene Universal Inc., by cloning the human DDB2 cDNA between BglII-XhoI sites of pmNeonGreen-C1 plasmid. DDB2(human)-mCherry was made by Gene Universal Inc., by removing the mNeonGreen sequence by digesting with AgeI and BglII and cloning the mCherry sequence between XhoI-HindIII sites of pmNeonGreen-DDB2 plasmid. DDB2(mouse)-

mCherry, GFP-DDB1(mouse) and GFP-Cul4A were provided by Dr. Wim Vermeulen[67,81]. OGG1-GFP was provided by Dr. A. Campalans[82]. DDB2-Flag was purchased from ORIGENE (RC200390). DDB2(K244E)-Flag mutant was made using the QuickChange II Mutagenesis kit (Agilent, #200523).

**siRNA transfections**. 40 nM siRNA was transfected using Lipofectamine 2000 (Thermo Fisher Scientific, #11668027) in serum-free DMEM, according to the manufacturer's instructions. Fresh complete media was added 4–6 h post transfection. Immunofluorescence and western blots were performed 48 h post transfection, unless specified otherwise.

siRNAs used: Control siRNA: siGENOME non-Targeting siRNA Pool #2 (Dharmacon D-001206-14-05); OGG1: siGENOME Human OGG1(Dharmacon M-005147-03-0005); DDB2: 5'-AACUAGGCUGCAAGACUU-3'; DDB1: 5'-AACGG CUGCGUGACCGGACAC -3'; Cul4A: 5'-GAAGAUUAACACGUGCUGGdTdT -3'; XPC: siGENOME Human XPC (SMARTpool, Dharmacon M-016040-01-00 05); CSB: 5'- GUG UGC AUG UGU CUU ACG A -3'.

**8-oxoG immunofluorescence**. 100,000 cells were plated on coverslips in 35 mm dishes. siRNAs were transiently transfected for the experiments. 48 h post transfection, cells were fixed for 8-oxoG staining using the Trevigen 8-oxoG antibody (#4354-MC-050). Briefly, cells were fixed with 1:1 MeOH, acetone for 20 min on ice and coverslips were allowed to air dry. Fixed cells were next treated with 0.05 N HCl for 5 min on ice. After washing cells three times with 1X PBS, coverslips were incubated with 100 µg/ml RNAse in 150 mM NaCl, 15 mM sodium citrate for 1 h at 37 °C. Next, cells were washed sequentially in 1X PBS, 35%, 50%, and 75% EtOH, for 3 min each. Cellular DNA was then denatured in situ with 0.15 N NaOH in 70% EtOH for 4 min. After washing briefly 2x with 1X PBS, 0.2 µg/ml Hoechst 33342 (Thermo fisher scientific, #H3570) in 1X PBS was used to stain DNA for 10 min. Coverslips were washed sequentially in 70% EtOH containing 4% v/v formaldehyde, 50% and 35% EtOH, and 1X PBS for 2 min each. Finally, coverslips were incubated in 5 µg/ml proteinase K in 20 mM Tris, 1 mM EDTA, pH 7.5 (TE) for 10 min at 37 °C, washed several times with 1X PBS and blocked with 1% BSA, 10% normal goat serum in 1X PBS, 1 h at RT. Cells were washed 3x with 1X PBS, and incubated with anti-8-hydroxyguanine antibody (1:250) diluted in 1X PBS containing 1% BSA, 0.01% Tween 20 at 4 °C O/N in a humidified chamber. Next day, cells were washed several times with 1X PBS containing 0.05% Tween 20 for 5 min each and incubated in fluorescent secondary antibody conjugate, Donkey anti-mouse Alexa488 (1:1000; Thermo Fisher Scientific #A21202) in 1X PBS containing 1% BSA, for 1 h in the dark, at room temperature. Finally, cells were washed several times with 1X PBS containing 0.05% Tween 20 and rinsed with deionized water before mounting with Prolong Diamond Anti-Fade (#P36970; Molecular Probes).

**Enzyme-based PFGE assay for 8-oxoG detection**. Cells were transfected with control or DDB2 siRNA. 48 h post transfection, cells were treated with KBrO₃ to introduce 8-oxoG and harvested 0, 2, or 4 h post treatment for genomic DNA extraction. Genomic DNA was isolated from cells using the QIAGEN Tip-20 according to the manufacturer's instructions with some modifications[83]. Additionally, 100 µM of butylated hydroxytoluene (Sigma; #W218405) and deferoxamine mesylate (Sigma; #D9522) were added to both lysis buffers to minimize background oxidation[47]. 0.5 µg of genomic DNA was treated with FPG (NEB, 2.7U/µg DNA) (NEB, #M0240L) for 2 h at 37 °C in 1X Cutsmart buffer (NEB) to covert 8-oxoG to single nucleotide gaps. Then single nucleotide gaps were converted to double strand breaks (DSBs) by treating with S1 nuclease (1U/ug DNA) (Thermo Fisher, #EN0321) in 1x S1 Nuclease buffer at 37 °C for 1 h. To confirm that sufficient enzyme was present in the reaction, a control experiment was done using 0.5 µg genomic DNA and a 100-fold molar excess of labeled 37-mer oligo containing 8-oxoG and incubated with FPG and S1 nuclease and incubated as mentioned above. Samples were run on a 10% sequencing gel to confirm that all of the oligo was incised (Supplementary Fig. 2e, f). After adding 6X loading dye, the DNA was resolved by pulse field gel electrophoresis (PFGE) on a 1% Certified Megabase Agarose gel (Biorad, #1613109) in 0.5 X TBE. Samples were electrophoresed at 14 °C and 6 V with a 1 s initial switch, and 6 s final switch for 17 h using a CHEF-DR II apparatus (BioRad). The gel was stained by ethidium bromide and imaged using a ChemiDoc (Biorad).

Weighted mean DNA lengths (MDL) were calculated using ImageQuant[48,49]. Briefly, a grid consisting of 30 rows was used on each lane, including the ladder. The grid in which each marker fell was noted, and the data were exported to Microsoft Excel. A standard curve was made using the $\log_{10}$(molecular size) from the ladder and the corresponding length of migration (exported from ImageQuant) and fitting a least square line. At least four points (between 2.5–12.5 kb) were used to generate the standard curve. This allowed for calculation of molecular size (kb) at all 30 rows of each lane and determine the MDL and lesion frequency using the below mentioned formula:

(1) Weighted mean DNA length (MDL) = Σ (MWi x Vi) / Σ(Vi) MWi: length of DNA at each row (kb); Vi: Integrated volume at each row
(2) Lesion frequency = (MDL untreated /MDL treated) -1
To determine 8-oxoG lesions repaired, the fold change in lesion removal was calculated from MDLs and plotted:

(3) Fold change = Frequency in experimental sample/ Frequency in control sample

Preliminary experiments with U2OS cells were done to determine 8-oxoG lesion frequency after treatment with 20 and 40 mM KBrO₃ for 1 h. Results obtained were similar to lesion frequencies reported in the literature (background levels: 1.6 8-oxoG/10⁶ bases, 20 mM KBrO₃: 3.5 8-oxoG/10⁶ bases, 40 mM KBrO₃: 7.4 8-oxoG/ 10⁶ bases)[40].

**Colony formation assay**. U2OS-FAP-TRF1 (WT and DDB2 KO or control and OGG1 KD) cells were plated in 6-well plates 24 h prior to treatment. The next day, cells were treated with KBrO₃ (Sigma 309087), (0–20 mM) for 1 h at 37 °C. After treatment, cells were trypsinized and counted, and 800 cells were plated in 6 cm dishes for the DDB2 KO experiment, and 500 cells were plated in each well of a 6-well plate for the OGG1 KD experiment (in triplicate for each condition). Cells were then allowed to recover for 8 days. On day 8, cells were fixed with 4% formaldehyde in PBS for 15 min at room temperature and colonies were stained using a 0.1% crystal violet, 20% methanol solution for 30 min at room temperature. The plates were washed with water and dried overnight before counting.

**Local UV-C damage**. Cells were washed with PBS. Using a 254 nm lamp, cells were exposed to 60 J/m² UV-C either globally or through a 2 µm polycarbonate filter (Millipore Sigma; #TTTP04700).

**Oxidative DNA damage generation**. 8-oxoguanine generation at telomeres: 100,000 cells were plated on coverslips in 35 mm dishes. 48 h post transfection, cells were incubated with 100 nM MG-2I dye for 15 min at 37 °C, 5% oxygen in phenol red-free DMEM. Cells were then exposed to 660 nm light (100 mW/cm²) for 10 min (unless specified otherwise) to induce the production of singlet oxygen. Cells were pretreated with transcription inhibitors for 90 min: α-amanitin (Sigma #A2263) and Cdk7 Inhibitor VIII, THZ1-Calbiochem (Sigma# 5323720001). Cells were fixed or harvested for further experiments.

Global 8-oxoguanine generation: Photosensitizer Ro 19-8022 was used to generate oxidative DNA damage (a kind gift from F. Hoffmann-La Roche, Ltd). Microscopic settings are explained in a separate section below. The following inhibitors were used: NEDD8 neddylation activating enzyme inhibitor (NAE1 inhibitor, MLN4924, Boston Biochem) and CSN5-catalysed cullin de-NEDDylation inhibitor (CSN5 inhibitor, SB-58-SN29, kindly provided by Novartis)[84].

**Immunofluorescence and fluorescence in situ hybridization (IF-FISH) to visualize recruitment of repair proteins at telomeres**. 100,000 cells were plated on coverslips in 35 mm dishes. Plasmids were transiently transfected for the experiments. 48 h post transfection, cells were treated with dye plus light, and allowed to recover for indicated time periods. Cells were incubated with ice-cold CSK buffer (100 mM NaCl, 3 mM MgCl2, 300 mM glucose, 10 mM Pipes pH 6.8, 0.5% Triton X-100) for 2 min before fixing with 4% paraformaldehyde for 10 min. Cells were washed thrice with PBS and permeabilized with 0.2% Triton X-100 for 10 min. After permeabilization, cells were blocked for 1 h at room temperature (10% goat serum, 1% BSA in PBS). Primary antibodies were added to the cells and incubated overnight at 4 °C. Next day, cells were washed thrice with PBS, and incubated with secondary antibodies for 1 h at room temperature. After three PBS washes, cells were fixed again with 4% paraformaldehyde for 10 min, washed in PBS, and dehydrated in 70%, 90%, and 100% ethanol for 5 min each. The hybridization solution (70% Di Formamide, 1× Maleic acid, 10 mM Tris, pH 7.5, 1× MgCl2, 0.1 µM PNA probe) was prepared and incubated at 85 °C for 3–5 min. PNA probes used: PNA Bio, F1004; (CCCTAA)3-Alexa488 or PNA Bio, F1013; (CCCTAA)3-Alexa647. After the coverslips dried, cells were hybridized for 10 min at 85 °C and incubated at room temperature for 2 h in a humid chamber, in the dark. After 2 h, coverslips were washed twice in hybridization wash buffer (70% formamide, 10 mM Tris-HCl pH 7.5) for 15 min each. Next, coverslips were washed thrice with PBS and incubated with DAPI (1:5000) for 10 min at room temperature. Finally, coverslips were washed once with PBS and dH₂O, and mounted on slides with Prolong Diamond Anti-Fade (#P36970; Molecular Probes).

Primary antibodies used: mCherry (1:250; Abcam #ab167453), GFP (1:100, Santa Cruz #B-2), Flag (1:500; CST #14793 S), TRF1 (1:500; abcam #10579). Secondary antibodies used: Donkey anti-mouse Alexa488 (1:1000; 1Thermo Fisher Scientific #A21202), Goat anti-Rabbit Alexa-594 (1:1000; Thermo Fisher Scientific #A11012).

**Quantification of protein colocalization at telomeres**. Images were acquired on the Nikon Ti inverted microscope with a 60X objective (1.4 NA) using a z stack of 0.2 µm. The exposure time of each channel was kept consistent throughout samples. Images were deconvoluted and analyzed using NIS Elements 5.2 advance research software.

For the quantification of foci, the region of interest (ROI) tool was used to label the nuclei. Next, in the measurement tab, a separate binary layer was created for the repair protein foci and the telomere foci. The intersection tool was then used to identify the third binary layer, which corresponded to the colocalized foci. The intensity threshold for each channel was kept consistent throughout the samples.

The foci counts were exported to Excel for analysis. The colocalized foci number was normalized to the telomere foci number of each nucleus to get the percent telomeres colocalized with the repair protein, which was reported.

**Proximity ligation assay**. 10,000 cells were plated in each well of an 8-chambered tissue culture treated glass slide (Falcon, #354118). Plasmids or siRNAs were transiently transfected for the experiments. 48 h post transfection, cells were treated with dye plus light, and allowed to recover for indicated time periods.

Cells were incubated with ice-cold CSK buffer for 2 min before fixing, permeabilizing and blocking as mentioned above. Primary antibodies mCherry (1:250; Abcam #ab167453), GFP (1:100; Santa Cruz #B-2) and TRF1 (1:500; abcam #10579) were added to the cells and incubated overnight at 4 °C.

Next day, probe incubation, ligation, and amplification were performed using the Sigma-Aldrich PLA kit (#DUO92101) according to the manufacturer's instructions. Images were acquired on the Nikon Ti inverted microscope with a 60X objective (1.4 NA) using a z stack of 0.2 μm. Images were deconvoluted and analyzed using NIS Elements 5.2 advance research software. PLA foci per nucleus was reported.

**Microscopic settings and local DNA damage induction using photosensitizer Ro 19-8022**. Cells were examined in normal culture medium and maintained at 37 °C and 5% CO2 within a large chamber included in the Leica TCS SP5 confocal microscope. Local DNA damage was induced in a sub-nuclear area with a diameter of 1.5 μm as described before[43]. For the induction of direct single strand breaks (SSBs) a 405 nm laser-pulse of 1 frame (2.595 s/frame) was used, corresponding to ~1 mW. For the induction of oxidative DNA damage, cells were first incubated for 10 min with 50 μM photosensitizer Ro 19-8022 and micro-irradiated as described above. The resulting accumulation curves were corrected for background values and normalized to the relative fluorescence signal before local irradiation. Data are presented as mean ± SEM from at least three independent, pooled experiments.

**Telomere volume measurements**. U2OS-FAP-TRF1 cells were imaged on a Sweptfield confocal system with a 1.2 pinhole at 100x magnification and a 1.5x coupler using a z stack of 0.13μm. RPE-FAP-TRF1 cells were imaged on the Nikon A1 confocal system using a 60x magnification, a pinhole of 1.2 and a z stack of 0.1 μm. All imaging conditions were kept consistent throughout samples. Images were deconvoluted using the Richardson Lucy method and the number and volumes of telomeres was analyzed using NIS Elements advance research GA3 software using a custom GA3 script.

**Immunoblotting**. Cell pellets were resuspended in 1X lysis buffer (Cell signaling #9803) containing 1 mM protease inhibitor (Millipore Sigma; #539134). Supernatants were obtained by centrifugation at 21,100 x g for 15 min at 4 °C. Protein was quantified using Bio-Rad protein assay (Bio-Rad, #5000006). Equal amounts of protein were diluted in 2X sample buffer (Bio-Rad; #1610737) and loaded on 4-20% tris-glycine polyacrylamide gels (Invitrogen; XP04202BOX). Proteins were transferred onto a polyvinylidene difluoride membrane and blocked in 20% nonfat dry milk (diluted in PBST: phosphate-buffered saline containing 0.1% Tween 20) for 1 h at room temperature. Membranes were incubated with primary antibodies for 2 h at room temperature or overnight at 4 °C. Membranes were washed 3 × 10 min in PSBT and incubated with peroxidase conjugated secondary antibodies for 1 h at room temperature. Membranes were washed again before developing using SuperSignal West Femto Maximum Sensitivity Substrate (Thermo Fisher Scientific; #34095). Primary antibodies used: DDB2 (1:1000; abcam #ab181136), OGG1(1:1000; abcam #124741), Cul4A (1:1000; CST #2699 S), DDB1 (1:1000; Invitrogen #37-6200), XPC (1:1000; CST #12701 S), CSB (1:1000; abcam #ab96089), mCherry (1:1000; Abcam #ab167453), β-actin (1:30,000; Sigma #A2228). Secondary antibodies used: anti-rabbit IgG (1:50,000 Sigma #A0545), or anti-mouse IgG (1:50,000 Sigma #A4416). Blots were analyzed on ImageJ v1.53k.

For Fig. 5e: Cells were collected in 2x sample buffer (125 mM Tris-HCl pH 6.8, 20% Glycerol, 10% 2-β-Mercaptoethanol, 4% SDS, 0.01% Bromophenol Blue), homogenized passing through a syringe tip and boiled at 98 °C for 5 min. Protein lysate was separated by SDS-PAGE and transferred to a PVDF membrane (0.45 μm, Merck Millipore). The membrane was blocked in 3% BSA and then incubated with primary and secondary antibodies for 2 h or overnight. Antibodies used were anti-DDB2 (ab181136, Abcam), anti-CUL4A (ab72548, Abcam), anti-CSA (ab137033, Abcam), anti-AQR (A302-547A, Bethyl Laboratories). Secondary antibodies were conjugated with CF IR Dye 680 or 770 (Sigma) and visualized using the Odyssey CLx Infrared Imaging System (LI-COR Biosciences).

**Quantification and statistical analysis**. Statistical analysis was performed as indicated in figure legends. Means of two groups were compared using two-tailed Student's *t*-test with a 95% confidence interval. Multiple comparisons were performed by one-way ANOVA or Two-way ANOVA with Sidak multiple comparisons test. All the analyses were performed on GraphPad Prism (V8.2) software.

**Reporting summary**. Further information on research design is available in the Nature Research Reporting Summary linked to this article.

## Data availability

The data that support this study are available from the corresponding author upon reasonable request. Source data are provided with this paper.

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

## Acknowledgements

We wish to thank Dr. Elise Fouquerel for the U2OS-FAP-TRF1 cell line. We thank Dr. Marcel P. Bruchez for providing the MG-2I and 660 nm light source. We thank Dr. Jacob Stuart-Ornstein, Dr. Roderick O'Sullivan, and Dr. Karen Arndt for helpful discussions. We also thank our lab members, Dr. Matt Schaich, Sripriya Raja, Dr. Wei Qian, Dr. Zhou Zhong and Brittani Schnable for careful reading of the manuscript. We are thankful to Dr. Jean Cadet for discussions regarding 8-oxoG lesion frequencies. Funding sources:

This work was supported by NIH grants, R01ES019566, R01ES028686, and R35ES031638 (B.V.H.), R35ES030396 (P.L.O.), F32AG067710 (R.P.B.), the gravitation program CancerGenomiCs.nl (H.L., A.P., and W.V.) and TOP-CW grant (714.017.003) (A.P. and W.V.) from the Netherlands Organization for Scientific Research. Oncode Institute is partly financed by the Dutch Cancer Society.

## Author contributions

B.V.H. and N.K. conceived the research. N.K. performed all immunofluorescence-FISH experiments. A.F.T performed all live-cell imaging experiments. W.V., H.L., and A.P. provided resources and support for live-cell imaging experiments. N.K. and Y.A. performed the PLA experiments. N.K. and V.R. performed all western blots. N.K., S.C.W., and M.C. performed the telomere volume analysis. R.P.B and P.L.O. provided RPE-FAP-TRF1 cell line. N.K. and B.V.H. drafted the paper, and B.V.H, N.K., W.V., H.L., A.F.T, A.P., R.P.B., and P.L.O. reviewed and edited the draft.

## Competing interests

The authors declare no competing interests.
