## [Peer Review File · Nature Communications]

Reviewers' Comments:

Reviewer #1:

Remarks to the Author:

The manuscript by Kumar et al. constitutes an extension of previous work from one of the authors' laboratory, whereby *in vitro* biochemical experiments suggested a novel role of DDB2 (generally known as a specific sensor of UV lesions that triggers nucleotide excision repair activity) in recognizing 8-oxoG (an oxidative DNA lesion) and recruiting the 8-oxoG DNA glycosylase. With the present manuscript, the authors attempt to demonstrate that this direct functional link between DDB2 and the base excision repair enzyme OGG1 also exists in the chromatin of living cells. The authors present a series of very elegant and sophisticated experiments that, in their view, provides evidence to conclude that DDB2 binds to 8-oxoG lesions to stimulate the downstream recruitment of OGG1 in human cells.

The experimental procedure involves a rather artificial oxidative burst to generate damage in telomeric DNA. By using this method, a high frequency of compound lesions is generated but the physiologic/pathologic relevance of the resulting DNA repair substrate is not clear. Even though 8-oxoG is considered the predominant type of damage, this oxidative burst at telomers may result (or is expected to result) in other types of base lesions, including cyclopurines as an example of nucleotide excision repair substrate. Therefore, I would be more cautious in concluding that the observed effects are due solely to the presence of 8-oxoG.

In vitro, the DDB1-DDB2 complex and/or the DDB2 subunit have been shown to recognize various forms of unusual DNA in addition to UV lesions, including certain cisplatin adducts as well as damage induced by nitrogen mustard and N-methyl-N'-nitro-N-nitrosoguanidine. The DDB1-DDB2 complex and/or the DDB2 subunit also detect abasic sites in DNA and bind to single-stranded DNA or to DNA containing a 2- or 3-base pair mismatch (Mutat. Res. 1994, 310:89; J. Biol. Chem. 2005, 280:39982). However, a biological significance of these findings for the cellular role of DDB1-DDB2 is not demonstrated. In view of these older findings, suggesting a general affinity of DDB2 for injured DNA, it is possible that the binding of DDB2 to 8-oxoG observed *in vitro* may have a limited *in vivo* relevance and that, in the experiments of this submitted manuscript, DDB2 rather binds to another of the compound lesions generated by the oxidative burst (there is more than just single-stranded breaks). Some of these additional lesions may constitute a substrate of nucleotide excision repair rather than base excision repair.

The authors of the manuscript should also consider that the observation that nucleotide excision repair factors may stimulate other repair pathways, including base excision repair, is not new and may be attributable to XPC, a key interaction partner of DDB2. In DNA-binding assays, the selectivity of XPC extends from bulky DNA lesions to include some non-bulky base modifications, including for example 8-oxoG or methyl-formamidopyrimidine moieties, that are typical substrates of the BER pathway (DNA Rep. 2013, 12:947; J. Cell Biol. 2012, 199:1037). Other previous reports demonstrated that XPC stimulates the activity of at least four distinct DNA glycosylases, including OGG1 (Mutat. Res. 2007, 614:37-47; Mutat. Res. 2011, 728:107). Moreover, mouse and human cells lacking functional XPC are hypersensitive to the cytotoxic effects of oxidative agents and also display an increased sensitivity to etoposide, a topoisomerase II inhibitor that causes DNA breaks (Cancer Res. 2007, 67:2567). These different lines of evidence indicate that it is actually XPC that serves as a general platform for the loading of multiple repair pathways, including base excision repair, to damaged DNA carrying compound lesions.

The final part of the manuscript attempting to link DDB2 to chromatin decompaction would need to be confirmed by a biochemical correlate of chromatin condensation like for example nuclease accessibility. I also wonder if it would be possible to carry out electron microscopy to visualize the more relaxed chromatin state.

Reviewer #2:

Remarks to the Author:

The results presented in this manuscript provide a step forward in our understanding of the interplay between the BER and NER pathways, for which, while there is extensive evidence, many mechanistic aspects remain to be uncovered. In particular, this work extends the finding of a potential role for DDB2 in the BER of 8-oxoG previously published by the authors. By inducing such lesions specifically in telomeres or throughout the whole genome, they show that DDB2 facilitates

the recruitment of OGG1 at the sites of damage. DDB2 is also required for XPC recruitment but not for that of XPA which is transcription-dependent. They also show that impairing DDB2 processing by CRLDDB2, leading to retention of the protein on densely damaged regions, interferes with the recruitment of OGG1. Finally, the results unveil a role for DDB2 in the decompaction of chromatin following the induction of oxidative lesions.

The first part of the paper (recruitment of DDB2 to telomeric lesions) recapitulates previous results from the same authors. The second part presents novel results showing the interdependence of the different BER and NER proteins for their recruitment and retention at the damaged sites. The approach used is the analysis of the co-localisation of the proteins with telomeres after damage induction by the chemoptogenetic approach developed by the Opresko laboratory. A third part uses live microscopy to analyse the recruitment of the proteins when oxidative lesions are induced by microirradiation in the presence of a photosensitizer.

While this study is well performed, it falls short of supporting some of the conclusions presented by the authors.

Main points to be addressed:

- 1) My main concern with this manuscript is that while all along the text, including the title, the authors link the observations and results presented to the mechanism of 8-oxoG repair, at no point they actually measure repair of the lesion to prove that DDB2 and the other NER proteins have an impact on 8-oxoG removal. While I appreciate that quantifying the 8-oxoG lesions induced by FAP at telomeres might be challenging, repair of the oxidised guanine induced by KBrO₃ can be followed by either mass spectrometry or immunofluorescence. Doing that on DDB2 KD or KO cells, together with the in vitro work previously published, would definitely prove the role of DDB2 in BER of 8-oxoG.
- 2) Related to the previous point, to link the sensitivity of the DDB2 KO cells to the deficiency in 8-oxoG repair, the authors should show that OGG1 KO cells are more sensitive to KBrO₃ than the "wild type".
- 3) The chemoptogenetic system of 8-oxoG induction used leading to one or two lesions per telomere and that according to the model a single DDB2 or OGG1 molecule is recruited per lesion, how can it be explained that DDB2 or OGG1 foci are detected? This point, together with the next one, should be addressed in the discussion.
- 4) The microirradiation experiments indicate that DDB2 is recruited to the damaged DNA before OGG1. However, OGG1 has > 10-fold affinity for 8-oxoG than DDB2 according to the published data: K_d < 8 nM for OGG1 (Auffret van der Kemp et al., doi: 10.1093/nar/gkh224) and > 100 nM for DDB2 (ref 45). How can this be reconciled with the model proposed by the authors?

Other comments:

It would be interesting to apply the microirradiation approach used in figure 5A-B to compare the recruitment kinetics of XPC with those of OGG1 and DDB2.

line 63: The reference for the role of APE1 in bypassing OGG1 AP-lyase activity is incorrect. The papers by Hazra et al. (doi: 10.1093/nar/29.2.430) and Vidal et al. (doi: 10.1093/nar/29.6.1285) should be cited.

line 371: It is not clear what this title means.

line 589: SV40-immortalized

line 589: the reference is not formatted

line 603: It is not mentioned 2.7 microliters of what

line 654: A more detailed description of the irradiation conditions (laser intensity, pulse duration) would be welcome.

Figure 1 H: KBrO₃ 3 in underscript

Reviewer #3:

Remarks to the Author:

This manuscript describes novel roles of UV-DDB (DDB1-DDB2 heterodimer) in repair of the major oxidative DNA base lesion 8-oxoG. The authors' group has previously reported that UV-DDB binds to DNA sites of 8-oxoG and stimulates activities of base excision repair (BER) proteins, such as OGG1 glycosylase and AP endonuclease 1. This was surprising and intriguing finding, because UV-DDB has been supposed to be involved in global nucleotide excision repair (NER) of UV-induced DNA damage, but not in BER.

Here the authors adopt a sophisticated chemoptogenetical approach to introduce 8-oxoG lesions specifically at telomeres. Using this system, they show that DDB2 as well as OGG1 is recruited to the telomeres upon induction of 8-oxoG. Depletion of OGG1 results in prolonged retention of DDB2 at telomeres presumably due to the presence of persistent lesions, whereas DDB2 knockdown compromised the recruitment of OGG1. Next the authors provide evidence that NER proteins, XPC and XPA, are also recruited to telomeres by induction of 8-oxoG. Although the recruitment of XPC depends on DDB2, roles for XPC in processing of 8-oxoG seems unclear. On the other hand, XPA is recruited in a DDB2-independent manner and probably involved in transcription-coupled BER. Then they focus on the DDB1-Cul4A-RBX1 ubiquitin ligase, which possibly regulate DDB2 turnover as well as chromatin structure after DNA damage. Surprisingly, depletion of DDB1 or Cul4A does not affect the recruitment of DDB2 at telomeres. Although both DDB1 and Cul4A themselves are recruited to telomeres, co-localization with DDB2 seems poor, suggesting their predominant roles in transcription-coupled repair over the DDB2-mediated 8-oxoG processing. The authors use another system to induce local 8-oxoG lesions at high density, which also confirms recruitment of both DDB2 and OGG1. Inhibitors affecting the CRL-DDB2 activities compromise the OGG1 recruitment, whereas the conclusion of this experiment is somewhat elusive. Finally, they provide with evidence that DDB2 is involved in decompaction of chromatin with 8-oxoG lesions at high density, which may increase accessibility of repair proteins and support efficient removal of the lesions.

This work proposes a novel molecular mechanism, by which detection and repair of 8-oxoG lesions are promoted in the context of chromatin. The manuscript is well written, and the experiments seem to have been performed mostly in high quality. Although this study possibly may extend our understanding of the mechanism underlying efficient 8-oxoG processing in vivo, some conclusions are not well supported by the presented data. This reviewer would like to raise concerns as follows, which should be addressed before considering publication.

1. The chemoptogenetical system inducing 8-oxoG lesions specifically at telomeres seems to be a powerful approach that can be highly evaluated. Although the authors show that some repair factors such as DDB2 and OGG1 are recruited to telomeres, there are some DDB2 and OGG1 foci clearly at non-telomeric sites. Especially in Fig. 2E and F, in the absence of DDB2, OGG1 foci increased over time, while co-localization with telomeres seems very poor. What do these non-telomeric foci mean?
2. Related to the above point. If the authors could visualize and quantify 8-oxoG lesions in cells, their conclusions would be strengthened substantially. This is probably possible with a specific antibody that is commercially available. Although immunofluorescence may give rise to quite some background signals due to endogenously generated 8-oxoG, this may provide with insights into the non-telomeric foci formation.
3. Fig. 1F and G: The authors show that the K244E mutation significantly impairs the recruitment of DDB2 to telomeres. This mutation has been known to make DDB2 totally deficient in binding to UV-damaged DNA, which is consistent with their data in Fig. S1F. However, the recruitment of DDB2 to telomeres was reduced only 2 fold (Fig. 1G). These results do not support the authors' statement: "DDB2 uses a similar damage recognition mechanism for UV-induced photoproducts and 8-oxoG" (line 164-165).
4. Fig. 3: The authors probably could show if depletion of XPC affects the recruitment of OGG1 to telomeres. In addition, if the XPA recruitment indeed depends on transcription-coupled repair, the conclusion could be strengthened substantially by showing the dependency on CSA and/or CSB.

5. Fig. 4: Previous studies (for instance, Rasic-Otrin et al. *Hum. Mol. Genet.* 12, 1507-1522, 2003) have shown that DDB2 deficient in interaction with DDB1 is unstable in vivo. The authors should show a relative expression level of DDB2-mCherry in comparison with endogenous DDB2, and if the siRNA targeting DDB1 affects expression of the individual DDB2 species. This is critical to confirm that the ineffectiveness of the DDB1 siRNA shown in Fig. 4B is not an artifact due to overexpression of DDB2-mCherry, for instance. It is also important to know which percentage of endogenous DDB2 is present in vivo as a DDB1-free form and if the level is relevant to 8-oxoG repair in general.

6. Related to point 5: Would it be possible to show by ChIP that endogenous DDB2 binds to telomeres without DDB1 in response to 8-oxoG induction?

7. Fig. 5: Here the authors use inhibitors of NAE and CSN5. Because these inhibitors affect all cullin-containing ubiquitin ligases, their effects must be highly pleiotropic. It is necessary to examine with this system if depletion of DDB2 or CSA affects the recruitment of OGG1.

RESPONSE TO REVIEWER COMMENTS

We greatly appreciate the thoughtful remarks of all three reviewers. We have completed nine new experiments to fully address their concerns as detailed below. All our responses are detailed below in red font.

Reviewer #1 (Remarks to the Author):

The manuscript by Kumar et al. constitutes an extension of previous work from one of the authors' laboratory, whereby in vitro biochemical experiments suggested a novel role of DDB2 (generally known as a specific sensor of UV lesions that triggers nucleotide excision repair activity) in recognizing 8-oxoG (an oxidative DNA lesion) and recruiting the 8-oxoG DNA glycosylase. With the present manuscript, the authors attempt to demonstrate that this direct functional link between DDB2 and the base excision repair enzyme OGG1 also exists in the chromatin of living cells. The authors present a series of very elegant and sophisticated experiments that, in their view, provides evidence to conclude that DDB2 binds to 8-oxoG lesions to stimulates the downstream recruitment of OGG1 in human cells.

The experimental procedure involves a rather artificial oxidative burst to generate damage in telomeric DNA. **By using this method, a high frequency of compound lesions is generated but the physiologic/pathologic relevance of the resulting DNA repair substrate is not clear.** Even though 8-oxoG is considered the predominant type of damage, **this oxidative burst at telomers may result (or is expected to result) in other types of base lesions, including cyclopurines as an example of nucleotide excision repair substrate. Therefore, I would be more cautious in concluding that the observed effects are due solely to the presence of 8-oxoG.**

We thank the reviewer for their concern regarding the FAP system and provide an explanation below:

We apologize for not making it clear that the FAP/MG-2I is a singlet oxygen specific photosensitizer when activated by near-infrared light (He et al. Nature Methods. 2016 Mar;13(3):263-268; PMID: 26808669). Jean Cadet's group used HPLC analysis to show that a significant amount of 8-oxoG is generated in the DNA when cells were incubated with a singlet oxygen generator (Ravanat et al. J Biol Chem. 2000 Dec 22;275(51):40601-4; PMID: 11007783), while levels of other lesions including thymidine glycols, 5-(hydroxymethyl)-2'-deoxyuridine, 5-formyl-2'-deoxyuridine, and 8-oxo-7,8-dihydro-2'-deoxyadenosine remained unchanged and at background levels. We have discussed these results with Jean Cadet, and he has indicated that singlet oxygen produces 8-oxoG exclusively.

We acknowledge that while the formation of other lesions is likely, especially during the process of repair (including single-strand and double-strand breaks), these lesions are not a direct result of FAP activation. Specifically, cyclopurine deoxynucleosides are not formed as a result of singlet oxygen generation (discussed in Brooks PJ. Neuroscience. 2007;145(4):1407-1417; PMID: 17184928). It is also known that cyclopurine deoxynucleosides are not substrates for BER (Brooks PJ et al. J Biol Chem. 2000 Jul 21;275(29):22355-62; PMID: 10801836).

Finally, the Opresko Lab has previously shown that dye plus light treatment in the FAP-TRF1 cell line generates lesions detectable by OGG1 (using immunofluorescence and enzyme-based assay for 8-oxoG detection) (Fouquerel E et al. Mol Cell. 2019 Jul 11;75(1):117-130; PMID: 31101499). We have also shown that OGG1 activity is stimulated by UV-DDB (Jang S et al. Nat Struct Mol Biol. 2019

Aug;26(8):695-703; PMID: 31332353) and that DDB2 facilitates recruitment of OGG1 using two different systems to introduce 8-oxoG (See Figure 2 and 5).

In vitro, the DDB1-DDB2 complex and/or the DDB2 subunit have been shown to recognize various forms of unusual DNA in addition to UV lesions, including certain cisplatin adducts as well as damage induced by nitrogen mustard and N-methyl-N'-nitro-N-nitrosoguanidine. The DDB1-DDB2 complex and/or the DDB2 subunit also detect abasic sites in DNA and bind to single-stranded DNA or to DNA containing a 2- or 3-base pair mismatch (Mutat. Res. 1994, 310:89; J. Biol. Chem. 2005, 280:39982). However, a biological significance of these findings for the cellular role of DDB1-DDB2 is not demonstrated. In view of these older findings, suggesting a general affinity of DDB2 for injured DNA, it is possible that the binding of DDB2 to 8-oxoG observed in vitro may have a limited in vivo relevance and that, in the experiments of this submitted manuscript, **DDB2 rather binds to another of the compound lesions generated by the oxidative burst (there is more than just single-stranded breaks). Some of these additional lesions may constitute a substrate of nucleotide excision repair rather than base excision repair.**

As we discussed in the response to the previous concern, singlet oxygen does not produce an "oxidative burst" per se, but a direct attack of singlet oxygen on the guanine base producing exclusively 8-oxoG. We thank the reviewer for raising the concern that binding of DDB2 to 8-oxoG may have a limited *in vivo* relevance.

To directly address the cellular relevance of DDB2 involvement in the removal of 8-oxoG we show that DDB2 KO or OGG1 KD cells are sensitive to KBrO₃ treatment, which causes primarily 8-oxoG damage (Ballmaier D, Epe B. Toxicology. 2006 Apr 17;221(2-3):166-71; PMID: 16490296). (See revised Figure 1E) (Jang S et al. Nat Struct Mol Biol. 2019 Aug;26(8):695-703; PMID: 31332353). Furthermore, we have performed two additional experiments: **a)** Examined 8-oxoG levels using immunofluorescence with antibodies to 8-oxoG in cells that either lack DDB2 or OGG1 and found an accumulation of endogenous 8-oxoG (See revised Figure 1A, B); **b)** Adapted and validated a FPG/S1 nuclease treatment (that produces double-strand breaks at 8-oxoG) combined with pulse-field gel electrophoresis assay to analyze KBrO₃-induced 8-oxoG formation and repair in high molecular weight genomic DNA. We show that a loss of DDB2 greatly reduces the rate of 8-oxoG removal (See revised Figure 1C, D and revised Supplementary Figure S2A-D).

The authors of the manuscript should also consider that the observation that nucleotide excision repair factors may stimulate other repair pathways, including base excision repair, is not new and may be attributable to XPC, a key interaction partner of DDB2. In DNA-binding assays, the selectivity of XPC extends from bulky DNA lesions to include some non-bulky base modifications, including for example 8-oxoG or methyl-formamidopyrimidine moieties, that are typical substrates of the BER pathway (DNA Rep. 2013, 12:947; J. Cell Biol. 2012, 199:1037). Other previous reports demonstrated that XPC stimulates the activity of at least four distinct DNA glycosylases, including OGG1 (Mutat. Res. 2007, 614:37-47; Mutat. Res. 2011, 728:107). Moreover, mouse and human cells lacking functional XPC are hypersensitive to the cytotoxic effects of oxidative agents and also display an increased sensitivity to etoposide, a topoisomerase II inhibitor that causes DNA breaks (Cancer Res. 2007, 67:2567). These different lines of evidence indicate that it is actually XPC that serves as a general platform for the

loading of multiple repair pathways, including base excision repair, to damaged DNA carrying compound lesions.

We completely agree with the reviewer regarding XPC's role in 8-oxoG removal and have referred to several previous studies in the manuscript. None of our data undermines these assertions. We also observed recruitment of XPC to telomeres after dye plus light treatment (Figure 3), which was dependent on DDB2, which has not been documented previously.

The final part of the manuscript attempting to link DDB2 to chromatin decompaction would need to be confirmed by a biochemical correlate of chromatin condensation like for example nuclease accessibility. I also wonder if it would be possible to carry out electron microscopy to visualize the more relaxed chromatin state.

Several studies provide evidence for the role of DDB2 in chromatin decompaction (Luijsterburg MS et al. *J Cell Biol.* 2012 Apr 16;197(2):267-81; PMID: 22492724; Lan L et al. *J Biol Chem.* 2012 Apr 6;287(15):12036-49. PMID: 22334663; Adam S et al. *Mol Cell.* 2016 Oct 6;64(1):65-78. PMID: 27642047) Furthermore, it was shown that DDB2 can shift the nucleosome register using structural studies on reconstituted nucleosomes (Matsumoto S et al. *Nature.* 2019 Jul;571(7763):79-84. PMID: 31142837). We have referred to these studies in our discussion section and review our results in the context of these supporting data. While we agree that additional experiments would be important in the future to examine DDB2's role in chromatin decompaction at 8-oxoG sites, it is beyond the scope of this present study.

Reviewer #2 (Remarks to the Author):

The results presented in this manuscript provide a step forward in our understanding of the interplay between the BER and NER pathways, for which, while there is extensive evidence, many mechanistic aspects remain to be uncovered. In particular, this work extends the finding of a potential role for DDB2 in the BER of 8-oxoG previously published by the authors. By inducing such lesions specifically in telomeres or throughout the whole genome, they show that DDB2 facilitates the recruitment of OGG1 at the sites of damage. DDB2 is also required for XPC recruitment but not for that of XPA which is transcription dependent. They also show that impairing DDB2 processing by CRL^{DDB2}, leading to retention of the protein on densely damaged regions, interferes with the recruitment of OGG1. Finally, the results unveil a role for DDB2 in the decompaction of chromatin following the induction of oxidative lesions.

The first part of the paper (recruitment of DDB2 to telomeric lesions) recapitulates previous results from the same authors. The second part presents novel results showing the interdependence of the different BER and NER proteins for their recruitment and retention at the damaged sites. The approach used is the analysis of the co-localisation of the proteins with telomeres after damage induction by the chemoptogenetic approach developed by the Opresko laboratory. A third part uses live microscopy to analyse the recruitment of the proteins when oxidative lesions are induced by microirradiation in the presence of a photosensitizer.

While this study is well performed, it falls short of supporting some of the conclusions presented by the authors.

We thank the reviewer for the kind remarks and comments and have addressed their concerns below.

Main points to be addressed:

1) My main concern with this manuscript is that while all along the text, including the title, the authors link the observations and results presented to the mechanism of 8-oxoG repair, **at no point they actually measure repair of the lesion to prove that DDB2 and the other NER proteins have an impact on 8-oxoG removal. While I appreciate that quantifying the 8-oxoG lesions induced by FAP at telomeres might be challenging, repair of the oxidised guanine induced by KBrO₃ can be followed by either mass spectrometry or immunofluorescence. Doing that on DDB2 KD or KO cells, together with the in vitro work previously published, would definitely prove the role of DDB2 in BER of 8-oxoG.**

We greatly appreciate the reviewer's thoughtful comment and have now two additional experiments to support the role of DDB2 in 8-oxoG repair, which has significantly improved the impact of our study:

- We performed an immunofluorescence for 8-oxoG and show that cells deficient in DDB2 or OGG1 had significant accumulation of 8-oxoG lesions (See revised Figure 1A, B). We also tried using immunofluorescence to study repair kinetics of 8-oxoG but had trouble getting consistent data as 8-oxoG signal was lost within 15 minutes after KBrO₃ treatment in control as well as DDB2/ OGG1 KD. Furthermore, we could not find a published study where 8-oxoG immunofluorescence was used to examine repair kinetics of 8-oxoG in the nucleus. Therefore, we used a different approach to track the repair.
- We used a repair enzyme-based assay combined with analysis of genomic DNA by pulse field gel electrophoresis to quantify 8-oxoG and show that absence of DDB2 significantly slows 8-oxoG repair following cellular treatment with KBrO₃ (See figure below; also see revised Figure 1C, 1D, Supplementary Figure S2A-D).

2) Related to the previous point, to link the sensitivity of the DDB2 KO cells to the deficiency in 8-oxoG repair, the authors should show that OGG1 KO cells are more sensitive to KBrO₃ than the “wild type”.

We thank the reviewer for bringing up this point. In response to this comment, we performed a colony formation assay in control and OGG1 knockdown cells after treatment with KBrO₃ and show increased sensitivity in OGG1-deficient cells (See revised Supplementary Figure S11).

3) The chemoptogenetic system of 8-oxoG induction used leading to one or two lesions per telomere and that according to the model a single DDB2 or OGG1 molecule is recruited per lesion, how can it be explained that DDB2 or OGG1 foci are detected? This point, together with the next one, should be addressed in the discussion.

The reviewer raises an important concern. The 8-oxoG lesion density at telomeres was estimated based on the enzyme-based assay combined with pulse field gel electrophoresis, assuming that the number of double-stranded breaks generated corresponds to the number of 8-oxoguanines in the telomeric DNA. (Fouquerel E et al. *Mol Cell*. 2019 Jul 11;75(1):117-130; PMID: 31101499). It is possible that that all 3 Gs of the TTAGGG sequence are converted to 8-oxoG after dye plus light treatment, which would look the same on a pulse field gel as when one 8-oxoG is present. The only way to differentiate between one 8-oxoG and tandem 8-oxoGs is by telomere sequencing. If there are multiple 8-oxoGs clustered, that could explain the foci formation.

Furthermore, in U2OS-TRF1-FAP cells, the FAP-TRF1 expression level is ~40% that of endogenous TRF1 (see figure below), therefore, the FAP-TRF1 protein binding could be heterogenous with some telomeres having more FAP bound than others which would generate more clustered damage in these telomeres leading to foci formation. Finally, it is also possible the DDB2 and OGG1 recruitment is enhanced in such a way that the local concentration of these proteins at sites of damaged telomeric chromatin leads to what looks like foci.

Western blot from Supplemental Figure S1 (Fouquerel E et al. *Mol Cell*. 2019 Jul 11;75(1):117-130; PMID: 31101499)

4) The microirradiation experiments indicate that DDB2 is recruited to the damaged DNA before OGG1. However, OGG1 has > 10-fold affinity for 8-oxoG than DDB2 according to the published data: K_d < 8 nM for OGG1 (Auffret van der Kemp et al., doi: 10.1093/nar/gkh224) and > 100 nM for DDB2 (ref 45). How can this be reconciled with the model proposed by the authors?

We agree with the reviewer that OGG1 has higher affinity for 8-oxoG than DDB2 when the lesion is in non-chromatin duplex DNA. However, recruitment of OGG1 and DDB2 to sites of damage are affected by both the concentration of the two proteins in the nucleus and the accessibility of 8-oxoG to the two proteins. When 8-oxoG is embedded in a nucleosome, OGG1 activity is greatly reduced (Olmon ED, Delaney S. *ACS Chem Biol*. 2017 Mar 17;12(3):692-701; PMID: 28085251) We hypothesize that DDB2

helps relax the chromatin to allow access to OGG1 when the damage is in the context of cellular chromatin. A recent finding showing that DDB2 can shift the nucleosome register on reconstituted nucleosomes (Matsumoto S et al. Nature. 2019 Jul;571(7763):79-84. PMID: 31142837) supports this hypothesis.

Other comments:

It would be interesting to apply the microirradiation approach used in figure 5A-B to compare the recruitment kinetics of XPC with those of OGG1 and DDB2.

We thank the reviewer for suggesting this experiment which has now been included in the manuscript (See revised Supplementary Figure S6A-B). Note that XPC is recruited more slowly than DDB2 supporting the hypothesis that DDB2 helps recruit XPC to sites of 8-oxoG repair.

line 63: The reference for the role of APE1 in bypassing OGG1 AP-lyase activity is incorrect. The papers by Hazra et al. (doi: 10.1093/nar/29.2.430) and Vidal et al. (doi: 10.1093/nar/29.6.1285) should be cited.

We thank the reviewer for pointing out our oversight. We have now cited the above-mentioned papers.

line 371: It is not clear what this title means.

We have changed the title to '*Timely removal of DDB2 from unrepaired 8-oxoG lesions requires CRL^{DDB2} mediated DDB2 dissociation*'. (Line 408)

line 589: SV40-immortalized

line 589: the reference is not formatted

line 603: It is not mentioned 2.7 microliters of what

line 654: A more detailed description of the irradiation conditions (laser intensity, pulse duration) would be welcome.

Imaging conditions were added to the Methods section (Line 741).

Figure 1 H: KBrO3 3 in underscript.

Thank you to the reviewer for noticing these minor errors in the text. We have reviewed the manuscript and corrected the errors accordingly.

Reviewer #3 (Remarks to the Author):

This manuscript describes novel roles of UV-DDB (DDB1-DDB2 heterodimer) in repair of the major oxidative DNA base lesion 8-oxoG. The authors' group has previously reported that UV-DDB binds to DNA sites of 8-oxoG and stimulates activities of base excision repair (BER) proteins, such as OGG1 glycosylase and AP endonuclease 1. This was surprising and intriguing finding, because UV-DDB has been supposed to be involved in global nucleotide excision repair (NER) of UV-induced DNA damage, but not in BER.

Here the authors adopt a sophisticated chemoptogenetical approach to introduce 8-oxoG lesions specifically at telomeres. Using this system, they show that DDB2 as well as OGG1 is recruited to the telomeres upon induction of 8-oxoG. Depletion of OGG1 results in prolonged retention of DDB2 at telomeres presumably due to the presence of persistent lesions, whereas DDB2 knockdown compromised the recruitment of OGG1. Next the authors provide evidence that NER proteins, XPC and XPA, are also recruited to telomeres by induction of 8-oxoG. Although the recruitment of XPC depends on DDB2, roles for XPC in processing of 8-oxoG seems unclear. On the other hand, XPA is recruited in a DDB2-independent manner and probably involved in transcription-coupled BER. Then they focus on the DDB1-Cul4A-RBX1 ubiquitin ligase, which possibly regulate DDB2 turnover as well as chromatin structure after DNA damage. Surprisingly, depletion of DDB1 or Cul4A does not affect the recruitment of DDB2 at telomeres.

Although both DDB1 and Cul4A themselves are recruited to telomeres, co-localization with DDB2 seems poor, suggesting their predominant roles in transcription-coupled repair over the DDB2-mediated 8-oxoG processing. The authors use another system to induce local 8-oxoG lesions at high density, which also confirms recruitment of both DDB2 and OGG1. Inhibitors affecting the CRL-DDB2 activities compromise the OGG1 recruitment, whereas the conclusion of this experiment is somewhat elusive. Finally, they provide with evidence that DDB2 is involved in decompaction of chromatin with 8-oxoG lesions at high density, which may increase accessibility of repair proteins and support efficient removal of the lesions.

This work proposes a novel molecular mechanism, by which detection and repair of 8-oxoG lesions are promoted in the context of chromatin. The manuscript is well written, and the experiments seem to have been performed mostly in high quality. Although this study possibly may extend our understanding of the mechanism underlying efficient 8-oxoG processing in vivo, some conclusions are not well supported by the presented data. This reviewer would like to raise concerns as follows, which should be addressed before considering publication.

We greatly appreciate the kind and thoughtful remarks by the reviewer and have responded to their concerns below. We believe that the comments and our experiments to mitigate these concerns have helped to significantly strengthen our findings providing a stronger and more impactful manuscript.

1. The chemoptogenetical system inducing 8-oxoG lesions specifically at telomeres seems to be a powerful approach that can be highly evaluated. Although the authors show that some repair factors such as DDB2 and OGG1 are recruited to telomeres, there are some DDB2 and OGG1 foci clearly at non-telomeric sites. Especially in Fig. 2E and F, in the absence of DDB2, OGG1 foci increased over time, while co-localization with telomeres seems very poor. What do these non-telomeric foci mean?

We wish to thank the reviewer for this observation. We reviewed the representative images and realized that an error was made when making the figure. The brightness was set higher for the DDB2

KD images so that the nucleus was visible in the FITC channel. We have changed the brightness/contrast so that it is consistent across all conditions and updated the figure (See revised Figure 2E). We apologize for this confusion and appreciate that the reviewer pointed it out.

Additionally, we quantified the fold increase in telomeric and non-telomeric OGG1-GFP foci compared to untreated in both control and DDB2 KD cells. We observed a significantly higher enrichment of OGG1-GFP at telomeres after dye plus light in controls but a greatly reduced enrichment of OGG1-GFP in DDB2 KD cells. We also observe a smaller but significant increase in non-telomeric foci over time, but we do not observe a particular trend when the controls and KDs were compared. One possible explanation for these non-telomeric foci is that it is possible that TRF1 can bind to TTAGGG sequences at some lower frequency throughout the genome.

2. Related to the above point. If the authors could visualize and quantify 8-oxoG lesions in cells, their conclusions would be strengthened substantially. This is probably possible with a specific antibody that is commercially available. Although immunofluorescence may give rise to quite some background signals due to endogenously generated 8-oxoG, this may provide with insights into the non-telomeric foci formation.

We appreciate the reviewer's thoughtful comment. As noted above in response to Reviewer 2, we have performed two additional experiments to support the role of DDB2 in 8-oxoG repair which we believe has significantly improved the manuscript:

- We performed an immunofluorescence for 8-oxoG and show that cells deficient in DDB2 or OGG1 had significant accumulation of unrepaired 8-oxoG (See revised Figure 1A, B).
- We used a repair enzyme-based assay to quantify 8-oxoG and show that absence of DDB2 slows 8-oxoG repair following cellular treatment with KBrO_3 (See revised Figure 1C, 1D, Supplementary Figure S2A-D).

3. Fig. 1F and G: The authors show that the K244E mutation significantly impairs the recruitment of DDB2 to telomeres. This mutation has been known to make DDB2 totally deficient in binding to UV-damaged DNA, which is consistent with their data in Fig. S1F. However, the recruitment of DDB2 to telomeres was reduced only 2-fold (Fig. 1G). These results do not support the authors' statement: "DDB2 uses a similar damage recognition mechanism for UV-induced photoproducts and 8-oxoG" (line 164-165).

While the recruitment of DDB2-K244E was reduced only by 2-fold compared to WT, the difference in DDB2-K244E enrichment between untreated and dye plus light treatment did not reach statistical significance and thus we stand by these results. We agree with the reviewer that additional experiments for example with purified proteins, would help strengthen our conclusion, but these experiments are beyond the scope of this current study and being pursued in a separate single-molecule study. We stand by our data, which strongly suggest that the K244 residue is important for damage detection and therefore, we have modified the results section accordingly (see line 192).

4. Fig. 3: The authors probably could show if depletion of XPC affects the recruitment of OGG1 to telomeres. In addition, if the XPA recruitment indeed depends on transcription-coupled repair, the conclusion could be strengthened substantially by showing the dependency on CSA and/or CSB.

We thank the reviewer for suggesting that we address these key points. We have included the recruitment of OGG1 in the absence of XPC and observed that OGG1 accumulation is unaffected by XPC knockdown (See revised Supplementary Figure S4C). Previous published data show that in human fibroblasts deficient for XPC, recruitment of OGG1 is not affected but dissociation of OGG1 is faster than in wild-type cells (Menoni H et al. J Cell Biol. 2012 Dec 24;199(7):1037-46. PMID: 23253478) It is possible that both XPC and OGG1 bind at the damaged site at the same time and are dependent on each other for stabilization, but this hypothesis will need to be tested systematically, probably using single-molecule techniques.

We also now have added new data showing that XPA recruitment to damaged telomeres is dependent on both CSB and OGG1, strengthening our conclusion that XPA participates in transcription-coupled repair of 8-oxoG (See revised Supplementary Figure S4D).

5. Fig. 4: Previous studies (for instance, Ropic-Otrin et al. Hum. Mol. Genet. 12, 1507-1522, 2003) have shown that DDB2 deficient in interaction with DDB1 is unstable in vivo. The authors should show a relative expression level of DDB2-mCherry in comparison with endogenous DDB2, and if the siRNA targeting DDB1 affects expression of the individual DDB2 species. This is critical to confirm that the inefficiveness of the DDB1 siRNA shown in Fig. 4B is not an artifact due to overexpression of DDB2-mCherry, for instance. It is also important to know which percentage of endogenous DDB2 is present in vivo as a DDB1-free form and if the level is relevant to 8-oxoG repair in general.

We thank the reviewer for highlighting this important study by Ropic-Otrin, and as suggested we looked at expression levels of endogenous and overexpressed DDB2 after transfection with DDB1 or Cul4A siRNA. While we observed that DDB1 KD had no effect on the levels of endogenous DDB2, we observed an apparent decrease in the overexpressed DDB2 species when DDB1 is knocked down, although the difference did not reach statistical significance. These data are now included in the revised manuscript (see revised Supplementary Figure S5D).

6. Related to point 5: Would it be possible to show by ChIP that endogenous DDB2 binds to telomeres without DDB1 in response to 8-oxoG induction?

We thank the reviewer for this suggestion. Instead of ChIP, we used proximity ligation assay to examine DDB2 recruitment to telomeres in the absence of DDB1. We observed that while DDB2 was efficiently being recruited to telomeric 8-oxoG in the absence of DDB1, there was a small reduction in recruitment as compared to controls (see revised Supplementary Figure S5C). We have discussed this result in the manuscript accordingly.

7. Fig. 5: Here the authors use inhibitors of NAE and CSN5. Because these inhibitors affect all cullin-containing ubiquitin ligases, their effects must be highly pleiotropic. It is necessary to examine with this system if depletion of DDB2 or CSA affects the recruitment of OGG1.

We thank the reviewer for bringing up this point. Unfortunately, CSA knockdown experiments are a bit complicated as CSA depletion reduces OGG1-GFP levels in these cells. It is therefore difficult to interpret such a KD experiment. Additionally, knocking down DDB2 had minimal effect on OGG1 recruitment in the photosensitizer plus 405nm light system. There are two differences between the FAP system and the photosensitizer plus 405nm light system:

1. The FAP is targeted to a specific region, therefore each time 8-oxoG is introduced in a relatively similar chromatin context. In the micro-irradiation system, a laser is used to introduce targeted damage at a random location in the cell.
2. The lesion density is much lower in the FAP-TRF1 system.

To directly address these concerns, we generated a cell line where FAP is fused to histone H2B and a fluorescent protein mCerulean (U2OS-H2B-FAP). For your review we have included preliminary data that once fully validated will be included in a new publication.

U2OS-H2B-FAP cells expressing FAP and mCerulean in the nucleus:

Cells were treated with different amounts of MG-2l dye (50nM, 100nM, 400nM) to vary the lesion density and exposed to 660nm light for 1 minute through a 1µm micropore membrane. The rationale is that increasing the dye concentration may increase the overall yield of 8-oxoG damage within a confined region at fixed amount of light.

Experimental setup for dye plus light treatment:

Cells were treated with MG-2l (indicated doses) for 15 minutes in phenol-red free media. Media was removed and a 1µm black membrane was placed on top. The dish was exposed to 660 nm light for 1 minute. Cells were fixed for immunofluorescence.

OGG1-GFP recruitment was examined by measuring the nuclear intensity in cells transfected with control or DDB2 siRNA, 30 minutes post dye plus light treatment. Data represents mean± SEM from 2-3 independent experiments. One-way ANOVA was performed: ****p<0.0001. We observed a significant decrease in OGG1-GFP intensity in DDB2 KD cells, suggesting that OGG1 recruitment depends on DDB2 even in non-telomeric regions.

OGG1-GFP recruitment in the presence or absence of DDB2 at lower MG-2l concentrations:

We then increased the dye concentration to 800nM. OGG1-GFP recruitment was examined by measuring the nuclear intensity in cells transfected with control or DDB2 siRNA, immediately after dye plus light treatment. Data represents one independent experiment. Student's t-test was performed: ns: not significant. We observed that OGG1 recruitment was no longer dependent on DDB2.

Complete work-up of this system will be presented in another publication, but we wanted to include these remarkable data here to help show the role that lesion density plays in DDB2- dependent OGG1 recruitment to 8-oxoG throughout the genome.

OGG1-GFP recruitment in the presence or absence of DDB2 at high MG-2I concentration:

These data show that with increasing levels of damage (by increasing the amount of the MG2I dye), the dependency of OGG1 recruitment on DDB2 is diminished.

Reviewers' Comments:

Reviewer #1:

Remarks to the Author:

There are no further comments

Reviewer #2:

Remarks to the Author:

The authors have responded adequately to all my comments and suggestions. I appreciate the effort made. By doing several new experiments, and in particular measuring the levels of 8-oxoG in the different conditions using two independent approaches and showing that the sensitivity to KBrO₃ can be related to the deficiency in repair of this lesion, I believe the manuscript is greatly strengthened.

Just a minor point concerning their response: Lebraud E et al., *Nucleic Acids Res.* 2020 Sep 18;48(16):9082-9097, doi: 10.1093/nar/gkaa611 used the immunofluorescence approach to assess the repair kinetics of 8-oxoG in the nucleus.

Reviewer #3:

Remarks to the Author:

This reviewer greatly appreciates the authors' sincere efforts to address the concerns.

1. The authors probably could further improve the representative images. For instance, in Fig. 2E, the OGG1-GFP signals now look too weak (at least on screen) and are difficult for readers to confirm the colocalization (in this sense, the merged images are not very effective).

3, 4. It seems appropriate to modify the authors' statements in such a way.

5, 6. Considering the presence of residual DDB1 after siRNA treatment and a much higher expression level of DDB1 compared to DDB2 (and other DCAF family members), it is still not fully convincing that DDB2 is recruited to telomeres as a DDB1-free form. They show that the siRNA targeting DDB1 reduced the level of DDB2-mCherry (but not endogenous DDB2), while this can be interpreted such that only DDB2 (regardless endo- or exogenously expressed) bound to DDB1 could survive. In the absence of direct evidence for the presence of DDB1-free DDB2, this reviewer thinks that the conclusions on this point should better be weakened substantially.

Fig. S5D, the blot on top of the middle column. Please indicate more clearly which antibody was used, mCherry or DDB2, for this.

Response to REVIEWERS' COMMENTS

Reviewer #1 (Remarks to the Author):

There are no further comments.

We thank the reviewer for taking the time to review our manuscript.

Reviewer #2 (Remarks to the Author):

The authors have responded adequately to all my comments and suggestions. I appreciate the effort made. By doing several new experiments, and in particular measuring the levels of 8-oxoG in the different conditions using two independent approaches and showing that the sensitivity to KBrO₃ can be related to the deficiency in repair of this lesion, I believe the manuscript is greatly strengthened.

Just a minor point concerning their response: Lebraud E et al., Nucleic Acids Res. 2020 Sep 18;48(16):9082-9097, doi: 10.1093/nar/gkaa611 used the immunofluorescence approach to assess the repair kinetics of 8-oxoG in the nucleus.

We thank the reviewer for pointing out the error and really appreciate their valuable feedback which has greatly strengthened the manuscript.

Reviewer #3 (Remarks to the Author):

This reviewer greatly appreciates the authors' sincere efforts to address the concerns.

We appreciate the reviewer's helpful comments which have significantly strengthened the manuscript.

1. The authors probably could further improve the representative images. For instance, in Fig. 2E, the OGG1-GFP signals now look too weak (at least on screen) and are difficult for readers to confirm the colocalization (in this sense, the merged images are not very effective).

We have adjusted the OGG1-GFP signal and hopefully it will look better on the screen now.

3, 4. It seems appropriate to modify the authors' statements in such a way.

We thank the reviewer for accepting the changes made in the manuscript.

5, 6. Considering the presence of residual DDB1 after siRNA treatment and a much higher expression level of DDB1 compared to DDB2 (and other DCAF family members), it is still not fully convincing that DDB2 is recruited to telomeres as a DDB1-free form. They show that the siRNA targeting DDB1 reduced the level of DDB2-mCherry (but not endogenous DDB2), while

this can be interpreted such that only DDB2 (regardless endo- or exogenously expressed) bound to DDB1 could survive. In the absence of direct evidence for the presence of DDB1-free DDB2, this reviewer thinks that the conclusions on this point should better be weakened substantially.

We greatly appreciate the reviewer bringing up this point. We have now shown by PLA that DDB2-mCherry is still recruited to damaged telomeres when DDB1 is knocked down (Fig S5C). Furthermore, we see very little colocalization between DDB2 and DDB1 after dye plus treatment (Fig 4D). The concept of DDB2 having a DDB1-independent role has previously been demonstrated in the context of UV damage (Adam et al, Mol Cell, 2016; PMID: 27642047).

Finally, we have used an optical trap based single-molecule approach to show that DDB2 can bind DNA independently of DDB1 and there is a dynamic interaction between these proteins during DNA repair. This study is part of a manuscript currently in preparation. This study supports our conclusion that DDB2 might not always be part of a DDB2-DDB1 heterodimer.

Fig. S5D, the blot on top of the middle column. Please indicate more clearly which antibody was used, mCherry or DDB2, for this.

The labelling has been added.